# CONTROLLING SPACE AND TIME WITH DIFFUSION MODELS

**Daniel Watson**[*], **Saurabh Saxena**[*], **Lala Li**[*], **Andrea Tagliasacchi, David Fleet**
Google DeepMind

## ABSTRACT

We present 4DiM, a cascaded diffusion model for 4D novel view synthesis (NVS), supporting generation with arbitrary camera trajectories and timestamps, in natural scenes, conditioned on one or more images. With a novel architecture and sampling procedure, we enable training on a mixture of 3D (with camera pose), 4D (pose+time) and video (time but no pose) data, which greatly improves generalization to unseen images and camera pose trajectories over prior works which generally operate in limited domains (e.g., object centric). 4DiM is the first-ever NVS method with intuitive metric-scale camera pose control enabled by our novel calibration pipeline for structure-from-motion-posed data. Experiments demonstrate that 4DiM outperforms prior 3D NVS models both in terms of image fidelity and pose alignment, while also enabling the generation of scene dynamics. 4DiM provides a general framework for a variety of tasks including single-image-to-3D, two-image-to-video (interpolation and extrapolation), and pose-conditioned video-to-video translation, which we illustrate qualitatively on a variety of scenes. See `https://4d-diffusion.github.io` for video samples.

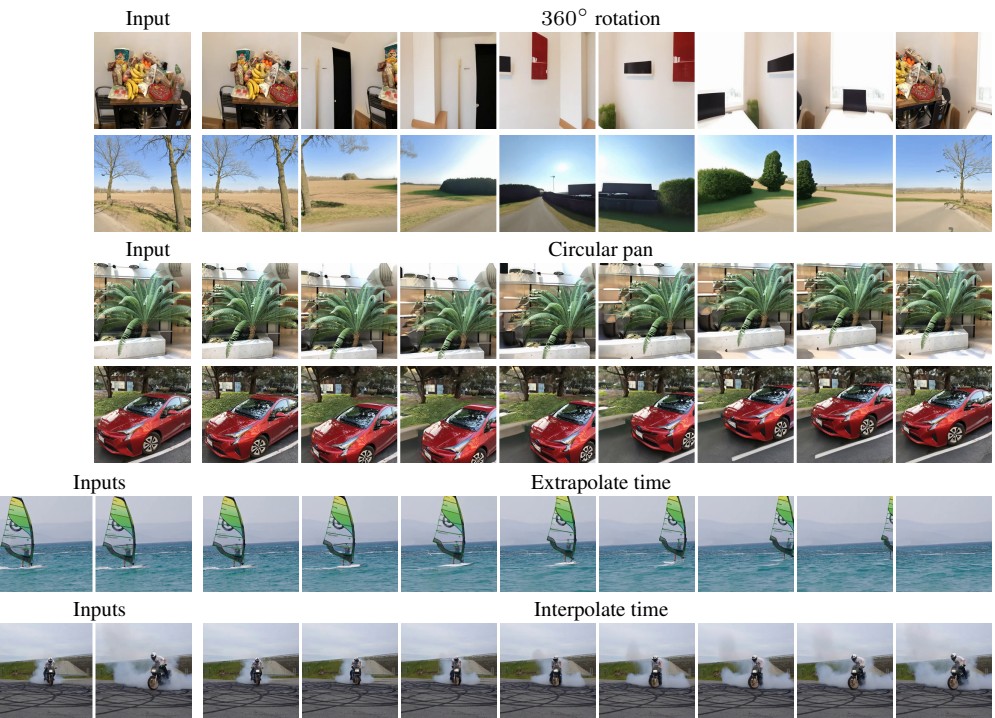

Figure 1: Zero-shot samples from 4DiM on LLFF (Mildenhall et al., 2019) and Davis (Pont-Tuset et al., 2017) given one or two input images. 4DiM excels at a wide variety of generative tasks involving camera and temporal control (e.g., free-form camera control, temporal interpolation and extrapolation). Note that 4DiM is a 32-frame model, so here we evenly sub-sample outputs. Samples are best seen as video: `https://4d-diffusion.github.io`.

---

[*]Equal contribution.

# 1 INTRODUCTION

4D generative models cast novel view synthesis (NVS) as a form of image-conditioned multi-view generation with control of camera pose and time. They extend the impressive capabilities of generative image and video models, leveraging large-scale learning to enable image generation with consistent 3D scene geometry and dynamics. As such, they promise to unlock myriad generative tasks such as novel view synthesis from a single image, image to 4D generation, image(s) to video, and video to video translation. Nevertheless, designing and training 4D models that generalize well to natural scenes, with accurate control over camera pose and time, remains challenging.

While there has been interest in diffusion models for NVS (Watson et al., 2022; Liu et al., 2023b), much of this work has focused on isolated objects with cameras located on the viewing sphere around the object. A key barrier to generalization, both to arbitrary scenes and free-form camera poses, has been the lack of rich, posed training data. The predominant datasets, i.e., RealEstate10K (Zhou et al., 2018) and CO3D (Reizenstein et al., 2021), are restricted to narrow domains, with camera poses estimated from structure-from-motion (SfM) that are noisy and lack metric scale information. Consequently, state-of-the-art 3D NVS models with good generalization to date have focused on somewhat specific domains (e.g., CAT3D (Gao et al., 2024) for object-centric scenes, or PNVS (Yu et al., 2023a) for indoor scenes), and they exhibit issues with pose alignment, where generated images do not coincide with the specified target viewpoints. In the case of 4D NVS, said challenges around lack of large-scale posed training data and quality of pose estimation are exacerbated. To this end, we advocate co-training with unposed video, as diverse video is available at scale and contains valuable information about the spatiotemporal regularities of the world.

To address these challenges, we introduce *4DiM*, a diffusion model for NVS conditioned on (one or more) images of arbitrary *scenes*, camera *pose*, and *time*. 4DiM extends previous 3D NVS diffusion models in three key respects: 1) from objects to *scenes*; 2) to free-form camera poses specified in meaningful physical units; and 3) with simultaneous control of camera pose and time. 4DiM is trained on a mixture of data sources, including posed 3D images/video and unposed video, of both indoor and outdoor scenes. We make this possible through an architecture with FiLM layers (Dumoulin et al., 2018) that default to the identity function for missing conditioning signals (e.g., videos without pose, or static scenes without time), which we call *Masked FiLM*, and through *multi-guidance*, which leverages this kind of training to adjust guidance weights for different conditioning signals separately. We also introduce calibrated versions of datasets posed via COLMAP (Schönberger & Frahm, 2016); calibrated data makes it easier to learn metric regularities in the world, like the typical sizes of everyday objects and spatial relations, and it enables one to specify camera poses in meaningful physical units for precise camera control. As we later show, training on poses with unknown scale will (unsurprisingly) make scale itself stochastic when generating samples given one input image, whereas fixing this problem yields both improved fidelity and pose alignment.

To summarize, our main contributions include:
- We introduce 4DiM, a pixel-based diffusion model for novel view synthesis conditioned on one or more images of arbitrary *scenes*, camera *pose*, and *time*. 4DiM comprises a base model that generates 32 images at $64 \times 64$ and a super resolution model that upsamples to $32 \times 256 \times 256$;
- We devise an effective data mixture for 4D models comprising posed and unposed video of indoor and outdoor scenes, yielding compelling results on camera control, time control, and promising zero-shot results controlling both simultaneously in video-to-video translation;
- We propose *Masked FiLM* layers to enable training with incomplete conditioning signals, and *multi-guidance* which leverages such training to further improve sample quality at inference time;
- We create a *scale-calibrated* version of RealEstate10K to improve model fidelity and enable metric pose control;
- We include extensive evaluation and comparisons to prior work, including fixes to SfM-based metrics (He et al., 2024), and a novel *keypoint distance* metric to detect the presence of dynamics.

# 2 RELATED WORK

Unlike the typical setting of Neural Radiance Fields (Mildenhall et al., 2021) (NeRF) where tens-to-hundreds of images are used as input for 3D *reconstruction*, pose-conditional diffusion models for NVS aim to extrapolate plausible, diverse, 3D consistent samples with as few as a single image input. Conditioning diffusion models on an image and relative camera pose was introduced by Watson et al.

(2022) and Liu et al. (2023b) as an effective alternative to prior few-view NVS methods, overcoming severe blur and floater artifacts (Sitzmann et al., 2019; Yu et al., 2021; Niemeyer et al., 2022; Sajjadi et al., 2022). However, they rely on neural architectures unsuitable for jointly modeling more than two views, and consequently require heuristics such as *stochastic conditioning* or Markovian sampling with a limited context window (Yu et al., 2023a), for which it is hard to maintain 3D consistency.

Subsequent work proposed attention mechanisms leveraging epipolar geometry to improve the 3D consistency of image-to-image diffusion models for NVS (Tseng et al., 2023), and more recently, finetuning text-to-video models with temporal attention layers or attention layers limited to overlapping regions (He et al., 2024; Wang et al., 2023a; Bahmani et al., 2024) to model the joint distribution of generated views and their camera extrinsics. These models however still suffer from several issues: they have difficulty with static scenes due to persistent dynamics from the underlying video models; they still suffer from 3D inconsistencies and low fidelity; and they exhibit poor generalization to out-of-distribution image inputs. Alternatives have been proposed to improve fidelity in multi-view diffusion models of 3D scenes, albeit sacrificing the ability to model free-form camera pose; e.g., see MVDiffusion and follow-ups (Tang et al., 2023; 2024) focusing on few, specific pose trajectories.

A concurrent body of related work on *3D extraction* has recently emerged following DreamFusion (Poole et al., 2022), where instead of directly training diffusion models for NVS, new techniques for sampling views parametrized as a NeRF with volume rendering are proposed such as *Score Distillation Sampling* (SDS) and *Variational Score Distillation* (Wang et al., 2024) (VSD). Here, a pre-existing diffusion model acts as a prior that drives the generation process. This enables, for example, text-to-3D using a text-to-image diffusion model. As demonstrated by ZeroNVS (Sargent et al., 2023), MVDream (Shi et al., 2023), ReconFusion (Wu et al., 2023) and CAT3D (Gao et al., 2024), using a pose-conditional diffusion model offers the advantage that the diffusion model can be conditioned on the sampled viewpoint for volume rendering during score distillation or NeRF postprocessing. Still, the extension of 3D extraction methods to 4D dynamic scenes and complex view-dependent reflectance phenomena remain challenging. In contrast, geometry-free diffusion models show encouraging results at scale with camera control, time control, and even their combination (Seitzer et al., 2023). Large-scale video models also provide encourage evidence of 3D geometrically consistent generation without an explicit rendering model (Brooks et al., 2024a; Li et al., 2024). Our success with 4DiM further supports this perspective, showing that one can simultaneously achieve fine-grained camera control and dynamics with a multiview diffusion model alone.

## 3 4D NOVEL VIEW SYNTHESIS MODELS FROM LIMITED DATA

4DiM uses a continuous-time diffusion model to learn a joint distribution over multiple views,

$$p(\boldsymbol{x}_{C+1:N} \,|\, \boldsymbol{x}_{1:C},\, \boldsymbol{p}_{1:N},\, t_{1:N})\,, \tag{1}$$

where $\boldsymbol{x}_{C+1:N}$ are generated images, $\boldsymbol{x}_{1:C}$ are conditioning images, $\boldsymbol{p}_{1:N}$ are relative camera poses (extrinsics and intrinsics) and $t_{1:N}$ are scalar, relative timestamps.[*] Following Ho et al. (2020), we choose our loss function to be the error between the predicted and actual noise ($L_{\text{simple}}$ in their paper), though we use the L1 rather than L2 norm, because, like prior work (Saharia et al., 2022a; Saxena et al., 2023), we found that it improves sample quality. Our models use the "$v$-parametrization" (Salimans & Ho, 2022), which helps stabilize training, and we adopt the noise schedules proposed by Kingma et al. (2021). Our current models process $N = 8$ or 32 images at resolution $256 \times 256$ ($N$ is the combined number of conditioning plus generated frames). To this end, we decompose the task into two models similar to prior work (Ho et al., 2022a;b); we first generate images at $64 \times 64$, and then up-sample using a $4\times$ super-resolution model trained with noise conditioning augmentation (Saharia et al., 2022c). We also finetuned the 32-frame model to condition on 2 and 8 frames to enable more comparisons and example applications of 4DiM models.

**Training data.** While 3D assets, multi-view image data, and 4D data are limited, video data is available at scale and contains rich information about the 3D world, despite not having camera poses. One of our key propositions is thus to train 4DiM on a large-scale dataset of 30M videos without pose annotations, jointly with 3D and 4D data. As shown in Section 5.1, video plays a key role in helping to regularize the model. The 3D datasets used to train 4DiM include ScanNet++ (Yeshwanth et al.,

---

[*]4DiM does not require sequential temporal ordering as in video models as the architecture is permutation-equivariant over frames. All $N$ images (conditioning and generated) are processed by the diffusion model.

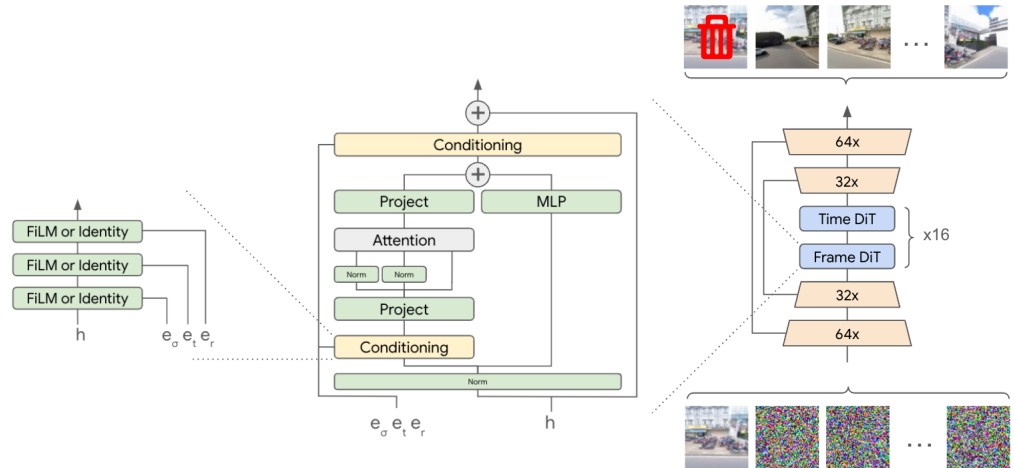

Figure 2: **Architecture** – We illustrate the 4DiM base model architecture, including choices for attention blocks and our conditioning mechanism. The 4DiM super-resolution model follows the same choices modulo minor differences to condition on low-resolution images. Outputs corresponding to the conditioning frames are always discarded from the training objective.

2023) and Matterport3D (Chang et al., 2017), which have *metric* scale, and more free-form camera poses compared to other common 3D datasets in the literature (e.g., CO3D (Reizenstein et al., 2021) and MVImgNet (Yu et al., 2023b)). We also use 1000 scenes from Street View with permission from Google, comprising posed panoramas with timestamps (i.e., it is a "4D" dataset). During training, we randomly sample views from Street View from the set of panorama images within $K$ consecutive timesteps ($K$=5 for our 8-view models, and $K$=20 for 32-view models). We sample unposed videos with probability 0.3 and views from posed datasets otherwise. 3D datasets are sampled in proportion to the number of scenes in each dataset.

**Calibrated RealEstate10K.** One particularly rich dataset for training 3D models is RealEstate10K (Zhou et al., 2018) (RE10K). It comprises 10,000 video segments of static scenes for which SfM (Schönberger & Frahm, 2016) has been used to infer per-frame camera pose, but only up to an unknown length scale. The lack of metric scale makes training more difficult because metric regularities of the world are lost, and it becomes more difficult for users to specify the target camera poses or camera motions in any intuitive units. This is especially problematic when conditioning on a single image, where scale itself otherwise becomes ambiguous. We therefore created a calibrated version of RealEstate10K by regressing the unknown metric scale from a monocular depth estimation model (Saxena et al., 2023). (For details, see Supplementary Material A.) The resulting dataset, *cRE10K*, has a major impact on model performance (see Section 4 below). We follow the same procedure to calibrate the LLFF dataset (Mildenhall et al., 2019) for evaluation purposes.

**Architecture.** Relatively little training data has both time and camera pose annotations. Most 3D data represent static scenes, while video data rarely include camera pose. Finding a way to effectively condition on both camera pose and time in a way that allows for incomplete training data is essential. We thus propose to chain *Masked FiLM* layers (Dumoulin et al., 2018) for (positional encodings of) diffusion noise levels, per-pixel ray origins and directions, and video timestamps. When any of these conditioning signals is missing (due to incomplete training data or random dropout for classifier-free guidance (Ho & Salimans, 2022)), the FiLM layers are designed to reduce to the identity function, rather than simply setting missing values to zero. This avoids the network from confusing a timestamp or ray coordinate of value zero with dropped or missing data. In practice, we replace the FiLM shift with zeros and scale with ones for masked signals. We illustrate the overall choices in Fig. 2. (For details see Supplementary Material C.)

**Sampling.** Steering 4DiM with the correct sampling hyperparameters is essential, especially the guidance weights for classifier-free guidance (Ho & Salimans, 2022) (CFG). In its usual formulation, CFG treats all conditioning variables as "one big variable". In practice, placing a different weight on each variable is important; e.g., text requires high guidance weights, but high guidance weights on images can quickly lead to unwanted artifacts. We thus propose *multi-guidance*, where we generalize CFG to do exactly this, without making independence assumptions between conditioning variables.

In essence, multi-guidance generalizes *2-guidance* originally proposed in InstructPix2Pix (Brooks et al., 2023) to $N$ variables. We start from a classifier-guided formulation (Dhariwal & Nichol, 2021), where we wish to sample with $k$ conditioning signals, $v_1, v_2, ...v_k$, using the score

$$\nabla_{\boldsymbol{x}} \log p(\boldsymbol{x}) + w_1 \nabla_{\boldsymbol{x}} \log p(v_1|\boldsymbol{x}) + \sum_{j=2...k} w_j \nabla_{\boldsymbol{x}} \log p(v_j|v_{1:j-1}, \boldsymbol{x}) . \qquad (2)$$

In the classifier-free formulation, this is equivalent to

$$(1 - w_1) \nabla_{\boldsymbol{x}} \log p(\boldsymbol{x}) + \sum_{j=2...k} (w_{j-1} - w_j) \nabla_{\boldsymbol{x}} \log p(\boldsymbol{x}|v_{1:j-1}) + w_k \nabla_{\boldsymbol{x}} \log p(\boldsymbol{x}|v_{1:k}) . \qquad (3)$$

If we train our model such that it drops out only $v_k$, or $v_{k-1}$ and $v_k$, ... , or all of $v_{1:k}$, we can sample with different guidance weights $w_i$ on each conditioning variable $v_i$. 4DiM is trained to drop out conditioning signals with probability 0.1, and it drops either $t_{1:N}$, $t_{1:N}$ and $\boldsymbol{p}_{1:N}$, or all of $t_{1:N}$, $\boldsymbol{p}_{1:N}$ and $\boldsymbol{x}_{1:C}$. This is a natural choice, since poses and timestamps without corresponding conditioning images do not convey useful information. For best results, 4DiM uses a guidance weight 1.25 on conditioning images and weights of 2.0 on camera poses and timestamps. We include an ablation study in our Supplementary Material B that demonstrates how multi-guidance provides additional knobs to push the Pareto frontiers between fidelity and dynamics / pose alingnment.

## 4 EVALUATION

Evaluating 4D generative models is challenging. Typically, methods for NVS are evaluated using generation quality. In addition, for pose-conditioned generation it is desirable to find metrics that capture 3D consistency (rendered views should be grounded in a consistent 3D scene) and pose alignment (the camera should move as expected). Furthermore, evaluating temporal conditioning requires some measure that captures the motion of dynamic content. We leverage several existing metrics and propose new ones to cover all these aspects, as described below.

**Image and Video fidelity.** FID (Heusel et al., 2017) is a key metric for image quality, but if used in isolation it can be uninformative and a poor objective for hyper-parameter selection. For example, Saharia et al. (2022b) found that FID scores for text-to-image models are optimal with low CFG weights (Ho & Salimans, 2022), but text-image alignment suffers under his setting. In what follows we report FID as well as the improved Frechet distance with DINOv2 features (FDD) (Oquab et al., 2023), which appears to correlate with human judgements better than InceptionV3 features (Szegedy et al., 2016). For video quality, we also report FVD scores (Unterthiner et al., 2018).

**3D consistency.** The work of Yu et al. (2023a) introduced the *thresholded symmetric epipolar distance* (TSED) to evaluate 3D consistency, as a more computationally efficient alternative to training separate NeRF models (Mildenhall et al., 2021) on every sample (e.g., like (Watson et al., 2022)). First, SIFT is used to obtain keypoints between a pair of views. For each keypoint in the first image, we can compute the epipolar line in the second image and its minimum distance to the corresponding keypoint. This is repeated for each keypoint in the second image to obtain a SED score. A hyperparameter threshold is selected, and the percentage of image pairs whose median SED is below the threshold is reported. Below, we report TSED with a threshold of 2.0 and include the TSED score of ground truth data, as poses in the data are noisy and do not achieve a perfect score.

**Pose alignment (revised metric: SfM distances).** We follow He et al. (2024) to quantify pose alignment, i.e., by running COLMAP pose estimation on the generated views, and comparing the camera extrinsics predicted by COLMAP to the target poses. To ensure the results are reasonable metrics, however, we modify the procedure of He et al. (2024) in several ways. First, due to the inherent scale and rotational ambiguity of COLMAP, we observe that it is necessary to align the estimated poses to the original ones before comparing their differences as described above. We also report the *relative* error in camera positions (**SfMD$_{\text{pos}}$**) and the angular deviation of camera rotations in radians (**SfMD$_{\text{rot}}$**) to account for scale variation across scenes. Finally, to get best results from COLMAP, we also feed it the input views. Please see our Supplementary Material E for more detail. We refer to this set of metrics as *SfM distances* hereon. Like we do for TSED scores, we also report the SfM distances for the ground truth data.

**Metric scale pose alignment.** The aforementioned metrics for 3D consistency and pose alignment are invariant to the scale of the camera poses as they rely on epipolar distances and SfM. In order to

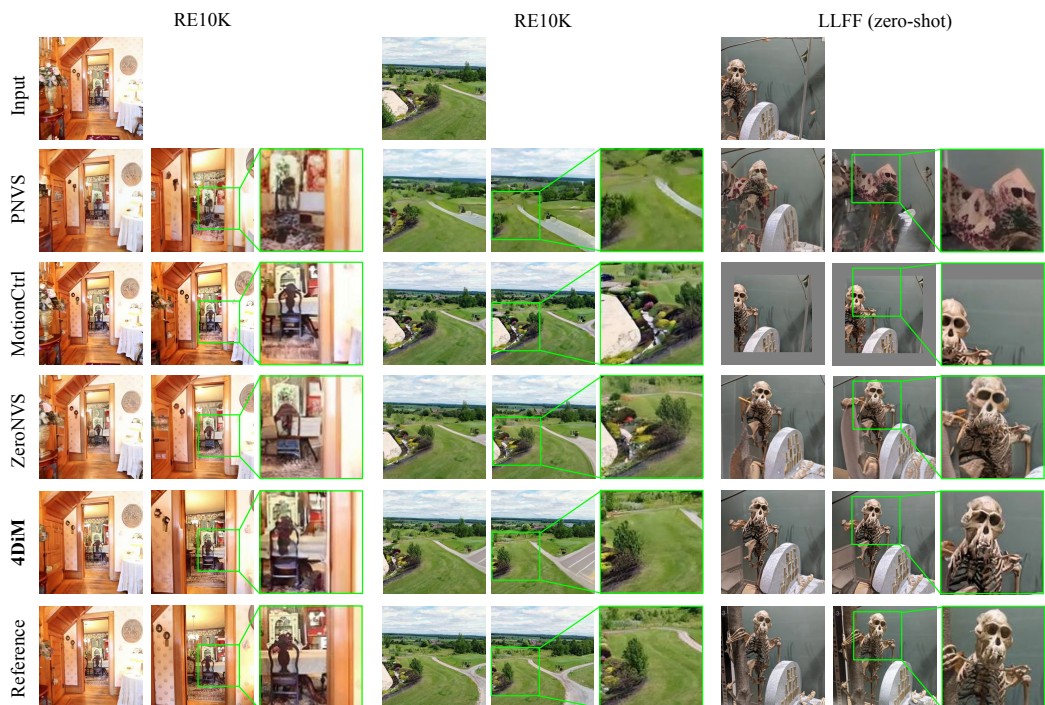

Figure 3: Qualitative comparison of 3D NVS across various diffusion models on in-distribution (RE10K) and out-of-distribution (LLFF) datasets. The model is an 8-frame 4DiM model trained solely on RE10K for 3D data for fair comparison with prior work (it is not our final 4DiM model). 4DiM demonstrates superior image fidelity and pose alignment compared to other methods, as can be seen in (1) the quality and realism of images furthest away from the input image, and (2) the location of objects and scene content alignment aligning with the ground truth images (shown in the bottom row). See our website for more samples: `https://4d-diffusion.github.io`.

evaluate the metric scale alignment of 4DiM we report PSNR, SSIM (Wang et al., 2004) and LPIPS (Zhang et al., 2018). These reconstruction-based metrics are not generally suitable to measure sample quality, as generative models can produce different but plausible samples. But for the same reason (i.e., that reconstruction metrics favor content alignment), we instead find them useful as an indirect indicators of *metric scale alignment*; i.e., given metric-scale poses, we would like models to not over/undershoot in position and rotation.

**Dynamics (new metric: keypoint distance).** One common failure mode that we observed early in our work is a tendency for models to copy input frames instead of generating temporal dynamics. Prior work has also noted that good FVD scores can be achieved with static videos (Ge et al., 2024). We therefore propose a new metric called *keypoint distance* (**KD**), where we compute the average motion of SIFT keypoints(Lowe, 2004) across image pairs with $k \geq 10$ matches. We report results on generated and reference views to assess whether generated images have a similar motion distribution.

## 5 EXPERIMENTS

We consider both in-distribution and OOD evaluations for 3D NVS in our ablations and comparisons to prior work. We use the RealEstate10K dataset (Zhou et al., 2018) as a common in-distribution evaluation. We use 1% of the dataset as our validation split and compute metrics on all baselines ourselves for this split, noting they might instead be advantaged as our test data may exist in their training data (for PNVS, all our test data is in fact part of their training dataset). For the OOD case, we use the LLFF dataset (Mildenhall et al., 2019). We present our main results on 3D novel view synthesis conditioned on a *single* image in Tab. 1 and Fig. 3, comparing the capabilities of our 4DiM cascade against PhotoConsistentNVS[*] (Yu et al., 2023a) (PNVS), MotionCtrl (Wang et al., 2023b),

---

[*]For PNVS, we follow Yu et al. (2023a) and use a Markovian sliding window for sampling, as they find it is the stronger than stochastic conditioning (Watson et al., 2022).

| | FID (↓) | FDD (↓) | TSED$_{t=2}$ (↑) | SfMD$_{pos}$(↓) | SfMD$_{rot}$(↓) | LPIPS (↓) | PSNR (↑) | SSIM (↑) |
|---|---|---|---|---|---|---|---|---|
| cRE10k test | | | | | | | | |
| PNVS | 51.41 | 1007. | 0.9961 (1.000) | **0.9773** (1.024) | 0.3068 (0.3638) | 0.3899 | 16.07 | 0.3878 |
| MotionCtrl | 43.07 | 370.6 | 0.4193 (1.000) | 1.027 (1.032) | **0.2549** (0.3634) | 0.5003 | 12.74 | 0.2668 |
| ZeroNVS | 37.35 | 522.3 | 0.6638 (1.000) | 1.049 (1.028) | 0.3323 (0.3579) | 0.3792 | 15.30 | 0.3617 |
| **4DiM-R** | **31.23** | **306.3** | **0.9974** (1.000) | 1.023 (1.075) | 0.3029 (0.3413) | **0.2630** | **18.09** | **0.5309** |
| **4DiM** | 31.96 | 314.9 | 0.9935 (1.000) | 1.008 (1.034) | 0.3326 (0.3488) | 0.3016 | 17.08 | 0.4628 |
| cLLFF zero-shot | | | | | | | | |
| PNVS | 185.4 | 2197. | 0.7235 (0.9962) | 0.9346 (0.8853) | 0.2213 (0.1857) | 0.5969 | 12.04 | 0.1311 |
| MotionCtrl | 106.0 | 531.5 | 0.1515 (0.9962) | 0.9148 (0.9415) | **0.1366** (0.2216) | 0.7016 | 9.722 | 0.06756 |
| ZeroNVS | 75.95 | 860.2 | 0.2205 (0.9962) | 0.9315 (0.8980) | 0.2015 (0.1947) | **0.5008** | **12.58** | **0.1578** |
| **4DiM-R** | **63.48** | **353.2** | **0.9659** (0.9962) | 0.9265 (0.8951) | 0.2011 (0.1934) | 0.5403 | 11.55 | 0.1444 |
| **4DiM** | 63.78 | 356.8 | 0.9167 (0.9962) | **0.9131** (**0.8960**) | 0.1838 (0.1943) | 0.5415 | 11.58 | 0.1408 |

Table 1: **Comparison to prior work in 3D NVS.** We compare 8-frame 4DiM models against prior work, with one model trained solely with cRE10K (4DiM-R) for a more fair comparison, and one model trained with our full dataset mixture (4DiM). We evaluate in-distribution and zero-shot on LLFF. For 3D consistency and pose alignment, we include scores on real views (in parenthesis) to indicate plausible upper bounds (which may be imperfect as poses are noisy and SfM may fail). We highlight the best result in bold, and underline results where models score better than real images. We do not report scores when SfM fails, denoted with n/a. Our models generally outperform baselines substantially across all metrics.

| | FID (↓) | FDD (↓) | TSED$_{t=2}$ (↑) | SfMD$_{pos}$(↓) | SfMD$_{rot}$(↓) | LPIPS (↓) | PSNR (↑) | SSIM (↑) |
|---|---|---|---|---|---|---|---|---|
| RE10k test | | | | | | | | |
| **4DiM-R** (RE10k) | 32.40 | 335.9 | **1.000** (1.000) | **1.021** (1.022) | 0.3508 (0.3737) | 0.3002 | 16.64 | 0.4704 |
| **4DiM-R** (cRE10k) | **31.23** | **306.3** | 0.9974 (1.000) | 1.023 (1.075) | **0.3029** (0.3413) | **0.2630** | **18.09** | **0.5309** |
| LLFF zero-shot | | | | | | | | |
| **4DiM-R** (RE10k) | 71.07 | 521.2 | 0.7727 (0.9962) | **0.9003** (0.8876) | 0.2108 (0.1915) | **0.4568** | **12.87** | **0.2016** |
| **4DiM-R** (cRE10k) | **63.48** | **353.2** | **0.9659** (0.9962) | 0.9265 (0.8951) | **0.2011** (0.1934) | 0.5403 | 11.55 | 0.1444 |
| ScanNet++ zero-shot | | | | | | | | |
| **4DiM-R** (RE10k) | 23.68 | 248.0 | 0.9512 (0.9815) | 1.885 (1.858) | **1.534** (1.520) | 0.1938 | 20.74 | 0.6716 |
| **4DiM-R** (cRE10k) | **22.89** | **243.2** | **0.9685** (0.9815) | **1.851** (1.917) | 1.541 (1.550) | **0.1809** | **21.25** | **0.6952** |

Table 2: **Ablation showing the advantage of using calibrated data (metric-scale camera poses).** We compare 8-frame 4DiM models trained on cRE10K vs uncalibrated RE10K. For RE10K and LLFF, models trained on cRE10K are tested on calibrated data, and models trained on uncalibrated RE10K are tested on uncalibrated data for a fair comparison. The exception is ScanNet++, which is already scale-calibrated. We find that training on cRE10K substantially improves in-domain and OOD performance both in terms of alignment and image quality. This is especially clear on reference-based image metrics such as PSNR, LPIPS, SSIM (with the exception of LLFF, where scores may be too low to be significant).

| | FID (↓) | FDD (↓) | TSED$_{t=2}$ (↑) | SfMD$_{pos}$(↓) | SfMD$_{rot}$(↓) | LPIPS (↓) | PSNR (↑) | SSIM (↑) |
|---|---|---|---|---|---|---|---|---|
| cRE10k test | | | | | | | | |
| **4DiM** (no video) | 32.45 | 326.5 | **0.9974** (1.000) | 1.023 (1.029) | 0.3356 (0.3692) | **0.2713** | **17.80** | **0.5209** |
| **4DiM** | **31.96** | **314.9** | 0.9935 (1.000) | **1.008** (1.034) | **0.3326** (0.3488) | 0.3016 | 17.08 | 0.4628 |
| ScanNet++ test | | | | | | | | |
| **4DiM** (no video) | 24.61 | 250.7 | 0.9615 (0.9815) | 1.895 (1.764) | **1.446** (1.431) | 0.1824 | 21.34 | 0.6969 |
| **4DiM** | **23.48** | **223.7** | n/a (0.9815) | **1.706** (1.889) | 1.550 (1.503) | 0.1965 | 20.67 | 0.6434 |
| cLLFF zero-shot | | | | | | | | |
| **4DiM** (no video) | 64.34 | 401.4 | **0.9886** (0.9962) | 0.9157 (0.9067) | 0.1872 (0.1965) | 0.5437 | 11.45 | **0.1430** |
| **4DiM** | **63.78** | **356.8** | 0.9167 (0.9962) | **0.9131** (0.8960) | **0.1838** (0.1943) | 0.5415 | 11.58 | 0.1408 |

Table 3: **Ablation showing the advantage of co-training with video data.** We train an identical 8-frame 4DiM model without video data to reveal the net-positive impact of co-training with video. Co-training with video data improves fidelity and generalization. Qualitatively, this is evident in 4DiM's ability to extrapolate realistic content.

and the diffusion model prepared by ZeroNVS (Sargent et al., 2023). These are the strongest 3D NVS diffusion models for natural scenes with code and checkpoints available. For LLFF, we load the input trajectory of views in order, subsample frames evenly, and then generate 7 views given a single input image. RealEstate10K trajectories are much longer, so following PNVS, we subsample with a stride of 10. We use the 7-view NVS task conditioned on a single image, in order to avoid weakening the baselines: MotionCtrl can only predict up to 14 frames, and PNVS performance degrades with the length of the sequence as it is an image-to-image model. We trained versions of 4DiM that process 8-frames to this end (as opposed to using the main 32-frame 4DiM model and subsampling the output) for a more apples-to-apples comparison. Quantitative metrics are computed on 128 scenes from our RealEstate10K test split, and on all 44 scenes in LLFF.

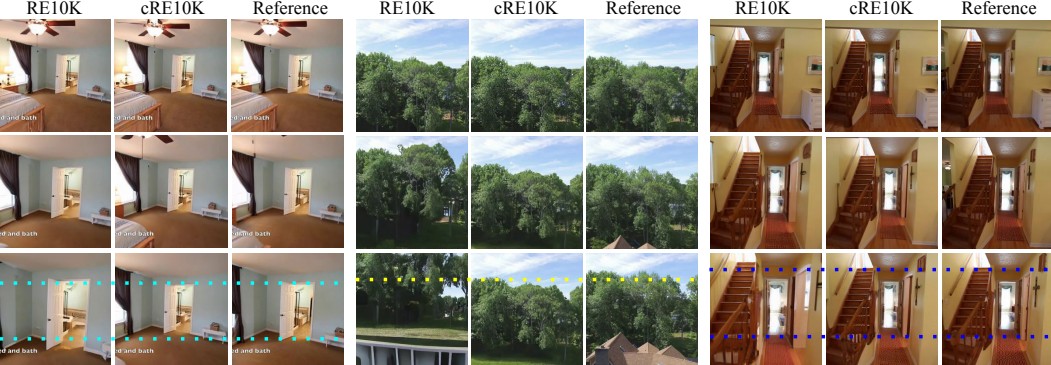

Figure 4: Qualitative comparison of 4DiM trained on calibrated vs uncalibrated RealEstate10K. The reference images show the scene that should be generated given the target camera pose. The model trained on calibrated data predicts scenes close to the ground truth, while the model trained on uncalibrated data often predicts scenes that have "overshot" along the trajectories. We include colored dotted lines on the furthest view from the conditioning input image to help illustrate the (mis)alignment of scene content (e.g., doorways, trees, stairs, etc.) of the generated samples when compared to the reference images.

We find that 4DiM achieves superior results in image quality (FID, FDD, FVD) by a wide margin compared to all the baselines, and generally outperforms them in reconstruction-based metrics (PSNR, SSIM, LPIPS) with the exception of ZeroNVS on the LLFF dataset. As much prior work has shown (Saharia et al., 2022c; Watson et al., 2022), reconstruction-based metrics penalize generated samples that may be plausible and tend to favor blurry images (for instance, diffusion architectures trained purely as regression models tend to outperform their generative counterparts in these metrics). Indeed, ZeroNVS generally produces samples with signficant blur (more qualitative comparisons are included in our Supplementary Material G). Qualitatively, we find that MotionCtrl does not suffer similarly from blur, but has trouble aligning with the conditioning target poses, as is evident in Figure 3. We also observe that PNVS exhibits more artifacts, with a substantial drop in quality in the zero-shot setting. Surprisingly, PNVS achieves the best TSED in LLFF but upon closer inspection, we find that the number of SIFT keypoint matches is ∼3x less than for 4DiM when computing TSED scores. This suggests potential improvements for TSED which we leave for future work, such as a higher threshold for the number of minimum keypoint matches, or requiring more spatial coverage of keypoints to discard examples for which the matches are too localized.

## 5.1 3D DATASETS SIGNIFICANTLY AFFECT NVS QUALITY

One of the most critical ingredients for 3D NVS, especially in the out-of-distribution setting, is the training dataset. We ablate this in three ways:

**More diverse 3D data is helpful.** Table 1 shows that adding more diverse 3D data beyond RealEstate10K to the training mixture improves pose alignment (SfM distances) on LLFF (zero-shot). The evaluation suggests a very small loss in fidelity when using the full mixture, which is expected since the mixture is more diverse and includes both indoor and outdoor scenes.

**Scale-calibrated data resolves ambiguity.** To determine the impact of consistent, metric scale training data, we compare a model trained with on the original RE10k data against one with calibrated metric scale poses. Neither model includes the other 3D data sources in their training data used to train our best 4DiM model. Quantitative results on 3D NVS from a single conditioning image are shown in Tab. 2 (including zero-shot performance on LLFF and ScanNet++), and qualitative results in Fig. 4. We find that models trained on uncalibrated data often overshoot or undershoot when scale is ambiguous, and that models trained on scale-calibrated data solve this problem.

**Large-scale video data improves generalization.** We also show the importance of video as a source of training data. Table 3 shows results comparing 8-frame 4DiM trained on our final data mixture to an otherwise identical model but trained without video. We find that including video data improves fidelity and generalization ability which is expected since available 3D/4D data is not as diverse. The improvement is significantly more apparent qualitatively (see Supplementary Material I for side-by-side samples), where it is clear that without video geometry becomes much worse.

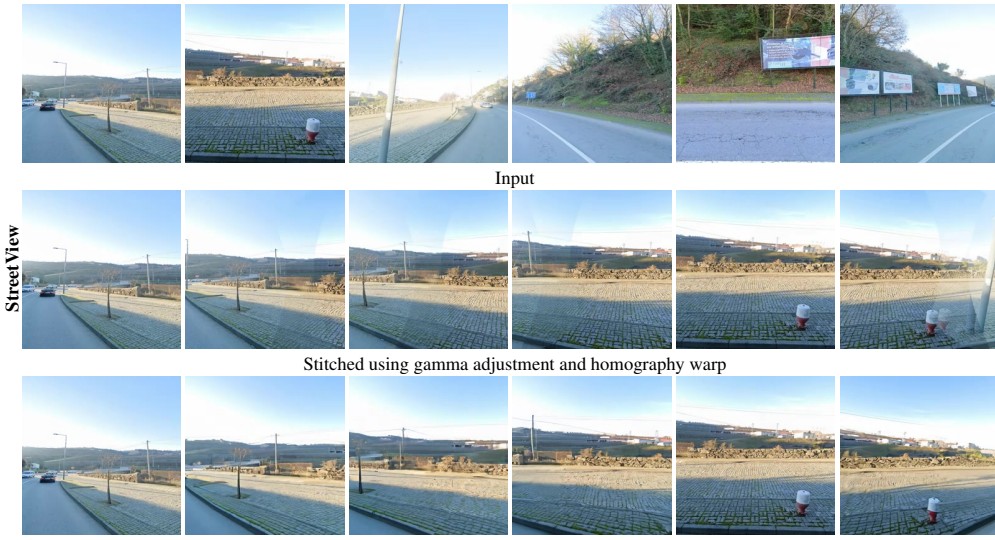

Figure 5: Stitching panoramas is non-trivial due to exposure differences, which is hard to fix with simple gamma/gain adjustment (middle row). Here we show novel views generated by 4DiM conditioned on 6 input frames covering a 360° field of view (bottom row). To use our 8-frame-input model, conditioning frames are randomly replicated to achieve the desired arity. (Note this is a subset of the 24 generated frames by 4DiM.)

|  | 30M video dataset test split | | | |
|---|---|---|---|---|
|  | FID ($\downarrow$) | FDD ($\downarrow$) | FVD ($\downarrow$) | KD ($\uparrow$) |
| **4DiM (1-frame conditioned)** | 62.73 | 355.4 | 934.0 | 2.214 (11.64) |
| **4DiM (2-frame conditioned)** | 56.92 | 334.7 | 809.8 | 2.820 (11.51) |
| **4DiM (8-frame conditioned)** | 43.34 | 320.7 | 400.2 | 3.181 (6.269) |

Table 4: **Comparing different numbers of input frames for video extrapolation**. With 32-frame 4DiM models, we find that increasing the number of conditioning frames greatly improves dynamics. Note that, because this is an extrapolation task with constant video frame rate, the true keypoint distance decreases with more conditioning frames (i.e., less frames to generate).

## 5.2 EMERGENCE OF TEMPORAL DYNAMICS

Early in our experiments, we observed that video from a single image can often yield samples with little motion. While this can be remedied by using more guidance weight on timestamps than on the conditioning images (see our Supplementary Material B), the single-image-conditioned case remains difficult. Nevertheless, with two or more input frames, 4DiM can successfully interpolate and extrapolate video and generate very rich dynamics. We quantitatively illustrate these results through our keypoint distance metric (along with other standard metrics) in Tab. 4. We also include quantitative and qualitative comparisons to prior frame interpolation methods in our Supplementary Material H. We suspect that future work including additional controls (e.g., text) on models like 4DiM, as well as larger scale, will likely play a key role in breaking modes for temporal dynamics, given that text+image-to-video has been successful for large models (Brooks et al., 2024b) and the same large-scale efforts report that at smaller scales dynamics quickly worsen (e.g., see SORA samples with 16x less compute).

## 5.3 MULTIFRAME CONDITIONING

While we (and most baselines) focus on single frame conditioning, using more than one frame unlocks interesting capabilities. Here we show two applications. First, we consider 4DiM on the task of panorama stitching (rendering novel viewpoints), given discrete images that collectively cover a full 360° FOV (see Fig. 5). For reference we stitch the images using a naive homography warp and gamma adjustment (Brown & Lowe, 2007). We find that outputs from 4DiM have higher fidelity than this baseline, and they avoid artifacts like those arising from incorrect exposure adjustment.

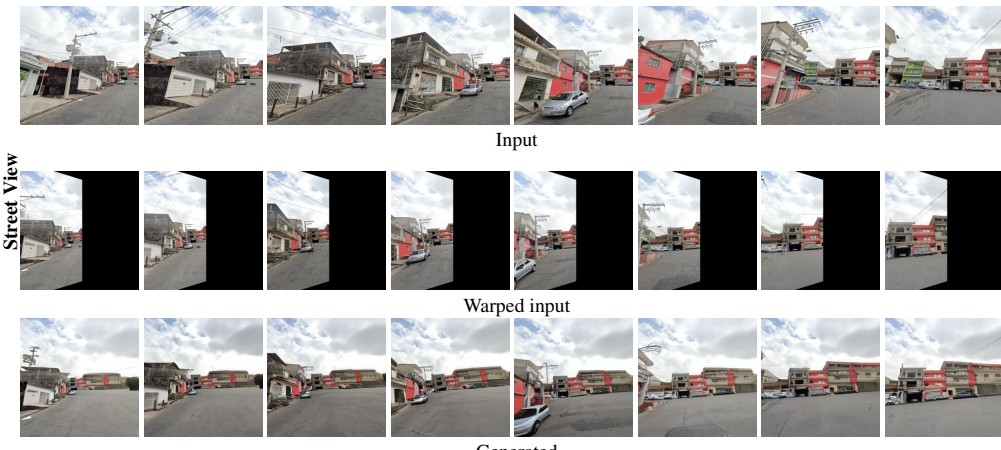

Figure 6: Novel views conditioned on a space-time trajectory on the Street View dataset. We condition on 8 consecutive time steps from the front-left camera and generate views at the same time steps with a camera facing the front. We find that the hallucinated regions (right half of the images) are consistent across space-time. Warped input images are provided for reference. We include more samples for video-to-video translation at `https://4d-diffusion.github.io`.

| # conditioning | Street View panoramas | | | | | Matterport3D 360° panoramas | | | | |
|---|---|---|---|---|---|---|---|---|---|---|
| frames | FID (↓) | FDD (↓) | PSNR (↑) | SSIM (↑) | LPIPS (↓) | FID (↓) | FDD (↓) | PSNR (↑) | SSIM (↑) | LPIPS (↓) |
| **1 (60°FOV)** | 119.7 | 1601 | 12.16 | 0.2925 | 0.5438 | 86.41 | 1080 | 9.366 | 0.1638 | 0.6467 |
| **6 (360°FOV)** | 37.43 | 354.1 | 17.76 | 0.5249 | 0.2663 | 24.86 | 239.4 | 15.64 | 0.4974 | 0.2987 |

Table 5: Quantitative performance on generating panos in the extrapolation (number of conditioning frames is 1) and interpolation (number of conditioning frames is 6) regime. Using more conditioning frames improves image fidelity which is to be expected. See Fig. 5 for samples.

We finally consider the task of rendering an existing space-time trajectory from novel viewpoints (i.e., pose-conditional video-to-video translation). In Fig. 6, we condition on images from a single camera at 8 consecutive frames from a Street View sequence and render different views yielding 3D-consistent hallucinated regions in the generated images. In our website (`https://4d-diffusion.github.io`), we include many more samples on video-to-video translation to demonstrate this works well in non-street scenes. We show that 4DiM can be used to stabilize the camera in an input video, and also move it along a novel trajectory even when the input video may feature a different camera motion.

# 6 DISCUSSION AND FUTURE WORK

To the best of our knowledge, 4DiM is the first model capable of generating multiple, approximately consistent views over simultaneous camera and time control from as few as a single input image. 4DiM achieves state-of-the-art pose alignment and much better generalization compared to prior work on 3D NVS. This new class of models opens the door to myriad downstream applications, some of which we demonstrate, e.g., changing camera pose in videos and seamlessly stitching panoramas. While still not perfect, 4DiM also achieves previously unseen quality in extremely challenging camera trajectories such as 360° rotation from a single image in the wild, indoor or outdoor.

**Limitations and future work.** Results with 4DiM are promising, but there is room for improvement, with more calibrated 3D/4D data and larger models, with which the improved capacity is expected to improve image fidelity, 360° camera rotation, and dynamics with single-frame conditioning.

**Societal impact.** Like all generative image and video models, it is important to develop and deploy models responsibly and with care. 4DiM is trained largely on data of scenes without people (or anonymized where present), and does not condition on text prompts, which mitigates many of the safety issues that might otherwise arise.

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

# Diffusion Models for 4D Novel View Synthesis
## Supplementary Material

This supplementary material expands upon the main paper, 'Diffusion Models for 4D Novel View Synthesis', by providing in-depth details and analyses. We first delve into the scale calibration process for the RealEstate10K dataset and the scene filtering methodology (Section A). We also discuss the effect of multi-guidance (Section B). Next, we provide a comprehensive overview of the 4DiM architecture, including its UViT backbone, transformer blocks, and camera ray conditioning (Section C). We then detail the training methodology and compute cost (Section D) and elaborate on the evaluation metrics used (Section E). Section F offers details of the implementation of baseline models from previous work, and Section G presents a qualitative comparison of 360° scene generation between 4DiM and ZeroNVS. Finally, we showcase the capabilities of 4DiM with an extensive collection of generated samples (Section J). For a more complete collection of samples, please also visit `https://4d-diffusion.github.io`.

## A    MORE DETAILS ON CALIBRATING AND FILTERING REALESTATE10K

To calibrate RealEstate10K, we predict a single length scale per scene with the help of a recent zero-shot model for monocular depth (Saxena et al., 2023) that predicts metric depth from RGB and FOV. For each image we compute metric depth using the FOV provided by COLMAP. The length-scale is then obtained by regressing the 3D points obtained by COLMAP to metric depth at image locations to which the COLMAP points project. An L1 loss provides robustness to outliers in the depth map estimated by the metric depth models. The mean and variance of the per-frame scales yields a per-sequence estimator. We use the variance as a simple measure of confidence to identify scenes for which the scale estimate may not be reliable, discarding ∼30% of scenes with the highest variance. Table 6 shows the importance of filtering out scenes with less reliable calibration. The calibrated dataset will be made public to facilitate quantitative comparisons.

| | FID (↓) | FDD (↓) | FVD (↓) | TSED$_{t=2}$ (↑) | SfMD$_{pos}$(↓) | SfMD$_{rot}$(↓) | LPIPS (↓) | PSNR (↑) | SSIM (↑) |
|---|---|---|---|---|---|---|---|---|---|
| cRE10k test | | | | | | | | | |
| **4DiM-R** (no filtering) | 31.81 | 313.8 | 237.6 | 0.9815 (1.000) | 1.035 (1.049) | 0.3328 (0.3529) | 0.3107 | 16.49 | 0.4465 |
| **4DiM-R** (w/ filtering) | **31.23** | **306.3** | **195.1** | **0.9974** (1.000) | **1.023** (1.075) | **0.3029** (0.3413) | **0.2630** | **18.09** | **0.5309** |
| ScanNet++ zero-shot | | | | | | | | | |
| **4DiM-R** (no filtering) | 24.35 | **232.0** | 143.1 | n/a (0.9815) | 1.881 (1.774) | 1.639 (1.512) | 0.1990 | 20.15 | 0.6436 |
| **4DiM-R** (w/ filtering) | **22.89** | 243.2 | **130.6** | 0.9685 (0.9815) | **1.851** (1.917) | **1.541** (1.550) | **0.1809** | **21.25** | **0.6952** |
| cLLFF zero-shot | | | | | | | | | |
| **4DiM-R** (no filtering) | **62.55** | 369.5 | 848.5 | 0.9167 (0.9962) | **0.9214** (0.8844) | 0.2194 (0.1932) | 0.5412 | 11.48 | 0.1389 |
| **4DiM-R** (w/ filtering) | 63.48 | **353.2** | **841.6** | **0.9659** (0.9962) | 0.9265 (0.8951) | **0.2011** (0.1934) | **0.5403** | **11.55** | **0.1444** |

Table 6: **Data filtering ablation.** We show the importance of discarding scenes where scale calibration is least reliable. Beyond the clear impact on fidelity, metric scale alignment itself improves by a wide margin. This is quantitatively shown through the reference-based image metrics (LPIPS, PSNR, SSIM) which favor content alignment, as discussed in Section 4.

## B    PUSHING PARETO FRONTS WITH MULTI-GUIDANCE

To assess the effectiveness of multi-guidance, we conducted a two-stage evaluation. First, we evaluated samples generated with various 'standard' guidance weights, where all conditioning variables (image, pose, and timestamp) were treated uniformly (illustrated in blue in Figure 7). Through qualitative and quantitative analysis, we identified the optimal standard guidance weight. Next, while holding this image guidance weight constant, we systematically varied the pose or timestamp guidance weights (shown in red). This procedure was applied to two tasks: single-image 3D generation using the cRE10K dataset, and video extrapolation from the first two frames using the DAVIS dataset. As demonstrated in Figure 7, our findings indicate that for video extrapolation, a slightly stronger timestamp guidance enhances performance in both dynamics and FID. For 3D generation, a slightly stronger pose guidance improves pose alignment (measured by TSED) with minimal impact on FID.

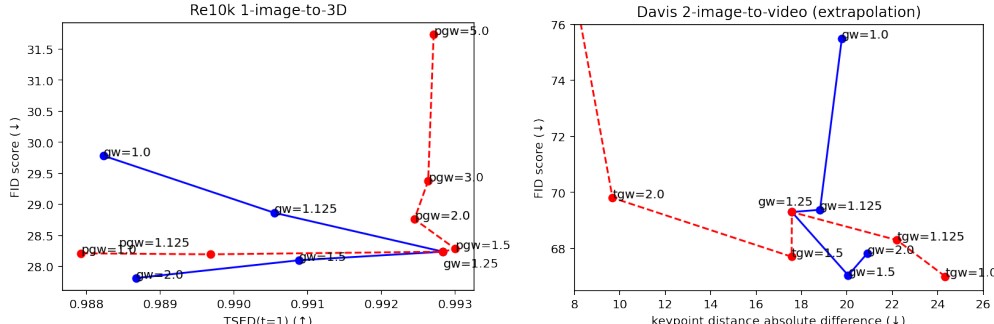

Figure 7: **Multi-guidance ablation.** Left: Pareto plot between FID and TSED (with threshold 1) for the single image to 3D task on the RealEstate10K dataset. Right: Pareto plot between FID and keypoint distance for the video extrapolation task from the two initial input frames on the Davis dataset. First, shown in blue, standard guidance, denoted with "gw", is swept. I.e., all conditioning variables have the same guidance weight. Then, shown in red, pose guidance (left) and timestamp guidance (right), denoted "pgw" and "tgw" respectively, are varied starting from the Pareto-optimal configuration in the initial sweep, leaving the guidance weights of other variables fixed. We find that overemphasizing pose guidance only helps marginally, likely because even without guidance the worst scores for pose alignment are already near-perfect ($\geq 98\%$ of keypoint matches are less than a pixel away from the epipolar line). And in the case of timestamp guidance, we find that this dramatically improves the presence of temporal dynamics in generated videos with a very small trade on FID score.

## C  Further details on neural architecture

**UViT backbone.** As hinted by Figure 2, 4DiM uses a UViT architecture following Hoogeboom et al. (2023). Compared to the commonly employed UNet architecture (Ronneberger et al., 2015), UViT uses a transformer backbone at the bottleneck resolution with no convolutions, leading to improved accelerator utilization. Information across frames mixes exclusively in temporal attention blocks, following (Ho et al., 2022b). We find that this is key to keep computational costs within reasonable limits compared to, say, 3D attention (Shi et al., 2023). Because the temporal attention blocks have a limited sequence length (32 frames), we insert them at all UViT resolutions. To mitigate the quadratic cost of attention, we use per-frame self-attention blocks only at the 16x16 bottleneck resolution where the sequence length is the product of the current resolution's dimensions. And to further minimize the amount of activations stored for backpropagation, we use only two UNet blocks when the resolution is at 32x or beyond, with hidden sizes (channels) decreasing in powers of two to a minimum of 256. Convolutional kernels at the 32x resolution have 3x3 kernels, and we increase the kernel size additively by 2 for higher resolutions to a maximum of 7x7. Our base model uses 16 transformer blocks at the bottleneck resolution of hidden size 128 with 16 heads, and our super-resolution model uses 8 transformer blocks of hidden size 128 with 8 heads.

**Transformer block design.** Our transformer blocks combine helpful choices from several prior work. We use parallel attention and MLP blocks following Dehghani et al. (2023), as we found in our early experiments that this leads to slightly better hardware utilization while achieving almost identical sample quality. We also inject conditioning information in a similar fashion to Peebles & Xie (2023), though we use fewer conditioning blocks due to the use of the parallel attention + MLP. We additionally emply query-key normalization following the findings of Gilmer et al. for improved training stability, but with RMSNorm (Zhang & Sennrich, 2019) for simplicity.

**Conditioning.** We use the same positional encodings for diffusion noise levels and relative timestamps as (Saharia et al., 2022b). For relative poses, we follow 3DiM (Watson et al., 2022), and condition generation with per-pixel ray origins and directions, as originally proposed by SRT (Sajjadi et al., 2022) in a regressive setting. Xiong et al. (2023) has compared this choice to conditioning via encoded extrinsics and focal lengths a-la Zero-1-to-3 (Liu et al., 2023b), finding it advantageous. While a thorough study of different encodings is missing in the literature, we hypothesize that rays are a natural choice as they encode camera intrinsics in a way that is independent of the target resolution, yet gives the network precise, pixel-level information about what contents of the scene are visible and which are outside the field of view and therefore require the model to extrapolate. Unlike

Plucker coordinates, which parametrize lines in 3D space, rays additionally preserve information about camera positions, which is key to deal with occlusions in the underlying 3D scene.

## D  FURTHER DETAILS ON COMPUTE, TRAINING TIME, AND SHARDING

We train 8-frame 4DiM models for ~1M steps. Using 64 TPU v5e chips, we achieve a throughput of approximately 1 step per second with a batch size of 128. All 8-frame models we train (i.e. base and super-resolution models) use this batch size, and we train all models with an Adam (Kingma & Ba, 2014) learning rate of 0.0001 linearly warmed up for the first 10,000 training steps, which we found was the best peak value in early sweeps. No learning rate decay is used. We follow Ho et al. (2020) and keep an exponential moving average of the parameters with a decay rate of 0.9999 to use at inference time for improved sample quality.

Available HBM is maximized with various strategies: first, we use bfloat16 activations (but still use float32 weights to avoid instabilities). We also use FSDP (i.e., zero-redundancy sharding (Rajbhandari et al., 2020) with delayed and rematerialized all-gathers). We then finetune this model to its final 32-frame version. This strategy allows us to leverage large batch-size pre-training, which is not possible with too many frames because the amount of activations stored for backpropagation scales linearly respect the number of frames per training example. The model is finetuned with the same number of chips, albeit only for 50,000 steps and at batch size of 32. This allows us to shard the frames of the video and use more than one chip per example at batch size 1. When using temporal attention, we simply all-gather the keys and values and keep the queries sharded over frames (one of the key insights from Ring Attention (Liu et al., 2023a)), though we don't decompose the computation of attention further as we find we have sufficient HBM to fully parallelize over all-gathered keys and values. FSDP is also enabled on the frame axis to maximize HBM savings so the number of shards is truly the number of chips (as opposed to the number of batch-parallel towers). This is possible because frame sharding requires an all-reduce by mean on the loss over the frame axis. Identically to zero-redundancy sharding, this can be broken down into a reduce-scatter followed by an all-gather.

## E  FURTHER DETAILS ON PROPOSED METRICS

We follow Yu et al. (2023a) and compute TSED only for contiguous pairs in each view trajectory, and discard sequences where we find less than 10 two-view keypoint matches. We use a threshold of 2.0 in all our experiments. For our proposed SfM distances, in order to get a scale-invariant metric, we first align the camera positions predicted by COLMAP by relativizing the predicted and original poses with respect to the first conditioning frame. This should resolve rotation ambiguity. Then, we resolve scale ambiguity by analytically solving for the least squares optimization problem that aligns the original positions to re-scaled COLMAP camera positions. Finally, we report the *relative error* in positions under the $L_2$ norm, i.e., we normalize by the norm of the original camera positions.

## F  FURTHER DETAILS ON BASELINES FROM PRIOR WORK

For PNVS, the conditioning image is the largest square center crop of the original conditioning frame, resized to $256 \times 256$ pixels. This is the same preprocessing procedure used by all our 4DiM models. PNVS generates output frames sequentially (one at a time), using 2000 denoising steps for each frame. All 4DiM samples are produced with 256 denoising steps.

For MotionCtrl, we use the checkpoint based on Stable Diffusion, which generates 14 frames conditioning on one image and 14 poses. We set the sampling hyper-parameter speed to 1, use 128 ddim denoising steps, and keep the other hyper-parameters as default values in the released script from MotionCtrl. To avoid any loss of quality due to an out-of-distribution resolution or aspect ratio, we use the resolution $576 \times 1024$, i.e. the same resolution as MotionCtrl was trained on. RE10K images already have the desired aspect ratio. LLFF images have resolution $756 \times 1008$; we are thus forced to crop them to $567 \times 1008$ so they have the same aspect ratio MotionCtrl was trained with. For comparability with 4DiM and PNVS, we *postprocess* MotionCtrl samples at $567 \times 1008$ by taking the largest center crop and then resizing to $256 \times 256$. Note that, in order to preserve the aspect ratio of LLFF, the resulting square outputs are for a smaller region than 4DiM and PNVS, hence the gray padding for MotionCtrl LLFF samples in Figure 3.

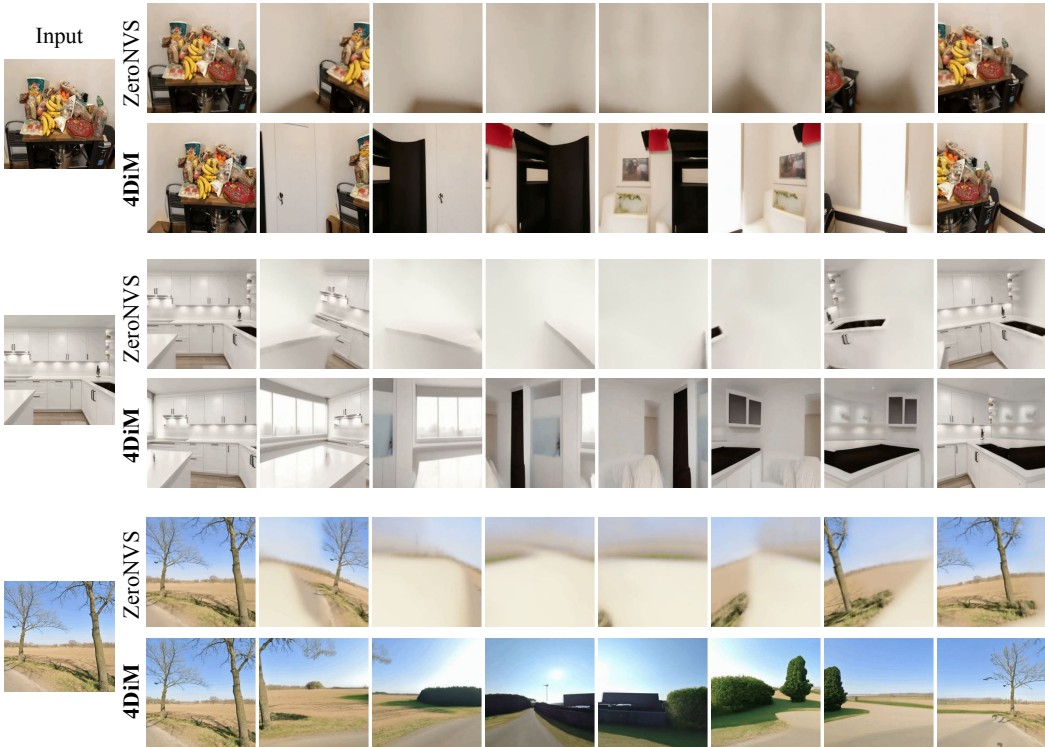

Figure 8: Qualitative comparison of 360° generation between 4DiM and ZeroNVS. 4DiM achieves superior sharpness and richer content.

For ZeroNVS, we use its diffusion model to evaluate novel view synthesis on RE10K and LLFF. Both input and output images are $256 \times 256$. We use default hyper-parameters as in the open sourced configuration, except the 'scene scale' parameter used for dealing with length scale ambiguity. We sweep over scale in $[0.7, 1.0, 1.2, 1.5, 2.0, 3.0, 5.0]$ and choose 3.0 for RE10K and 1.5 for LLFF.

## G  QUALITATIVE COMPARISON OF 360° GENERATION VS ZERONVS

We employ the ZeroNVS pipeline with score distillation sampling (SDS) to generate 360° views conditioned on a single image. For each scene, we use the default scene-generation configuration as open sourced by ZeroNVS, as adjusting *camera_distance* and *elevation_angle* yield worse and incoherent generation. We adjust the *scale* parameter for each scene, and use $1.0, 3.0, 3.0$ for the three scenes in Figure 8. Due to the computationally intensive nature of 360° generation with ZeroNVS (2-3 hours per scene), exhaustive hyperparameter tuning is impractical. We also observe another limitation: volume rendering will fail if the camera positions are not allowed to vary (or even if *camera_distance* is too small, e.g., 0.5), i.e., it is not feasible to keep the camera position fixed and only employ rotation. This occurs because there is no parallax and it is therefore not possible to infer depth/density (rotation alone only warps pixels). Figure 8 presents a comparison between 4DiM and ZeroNVS outputs, showing the superior sharpness and richer content achieved by 4DiM.

## H  COMPARISONS OF 4DiM VS PRIOR FRAME INTERPOLATION METHODS

We compare 4DiM against prior works on the task of frame interpolation (generating a video with large amounts of motion given the first and last frames). We use the Davis dataset as the benchmark and consider a generative baseline, LDMVFI (Danier et al., 2024), as well as the well-known (but non-generative) works FILM (Reda et al., 2022) and RIFE (Huang et al., 2022).

Below, we report FID scores of the middle frame for all methods, in order to consider the most challenging outputs and evaluate with the same number of samples despite that 4DiM generates 4x

more frames. We take the immediately previous frame to the middle frame in the case where a model generates an even number of frames.

| Method | FID |
|---|---|
| RIFE | 57.68 |
| FILM | 68.88 |
| LDMVFI | 56.28 |
| 4DiM | **52.89** |

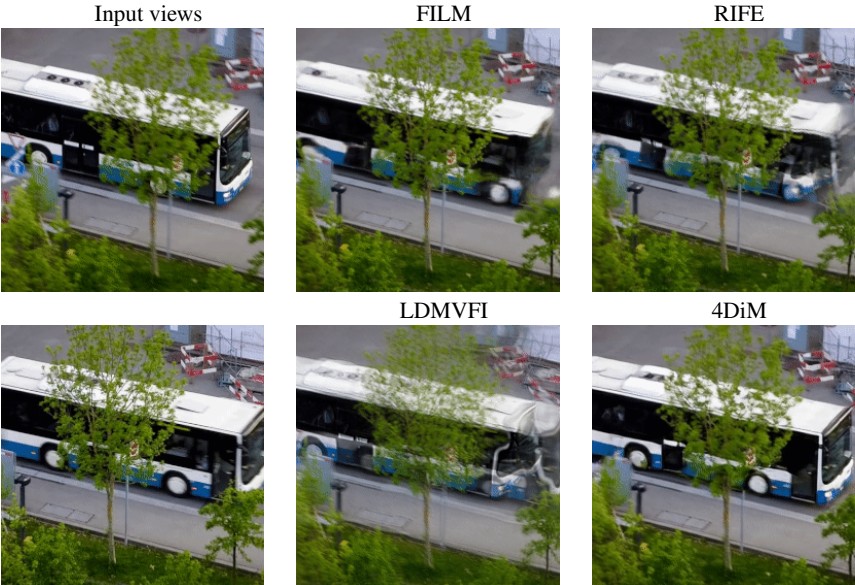

Figure 9: Comparison against frame interpolation baselines.

Qualitatively (Fig. 9), we find that 4DiM is competitive against all of these baselines, and does a much better job when the amount of motion is large. Non-generative baselines suffer from severe artifacts in these cases.

## I    ABLATION FOR CO-TRAINING WITH VIDEO DATA

In our main evaluations, we find that 4DiM trained without video performs slightly better on in-distribution evaluations (e.g., RealEstate10K). This is not too surprising as much of the capacity of the model now needs to be spent to model a more diverse data distribution. Still, to validate our claim that co-training with video is of paramount importance, it is necessary to consider out-of-distribution qualitative comparisons. Here it becomes evident that a model trained without video performs much worse. In particular, such a model does not produce plausible geometry for unknown objects (Fig. 10), whereas the model trained with large-scale video data does not suffer from this issue.

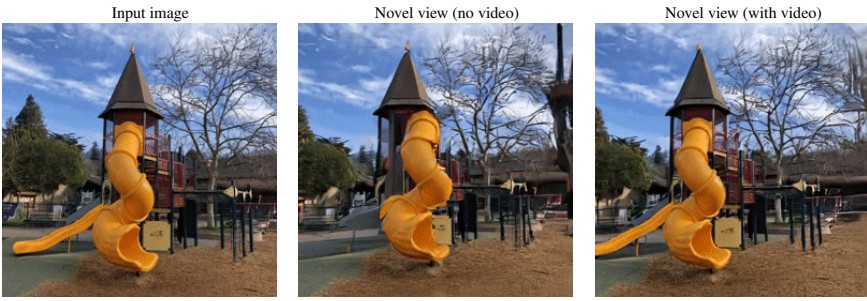

Figure 10: **Ablation for co-training with video.** We find this is key for geometric understanding in the wild. The model trained w/o video data infers incorrect geometry frequently in the LLFF dataset.

## J  ADDITIONAL 4DiM SAMPLES

We include more 4DiM samples below, though we strongly encourage the reader to
browse our website: https://4d-diffusion.github.io

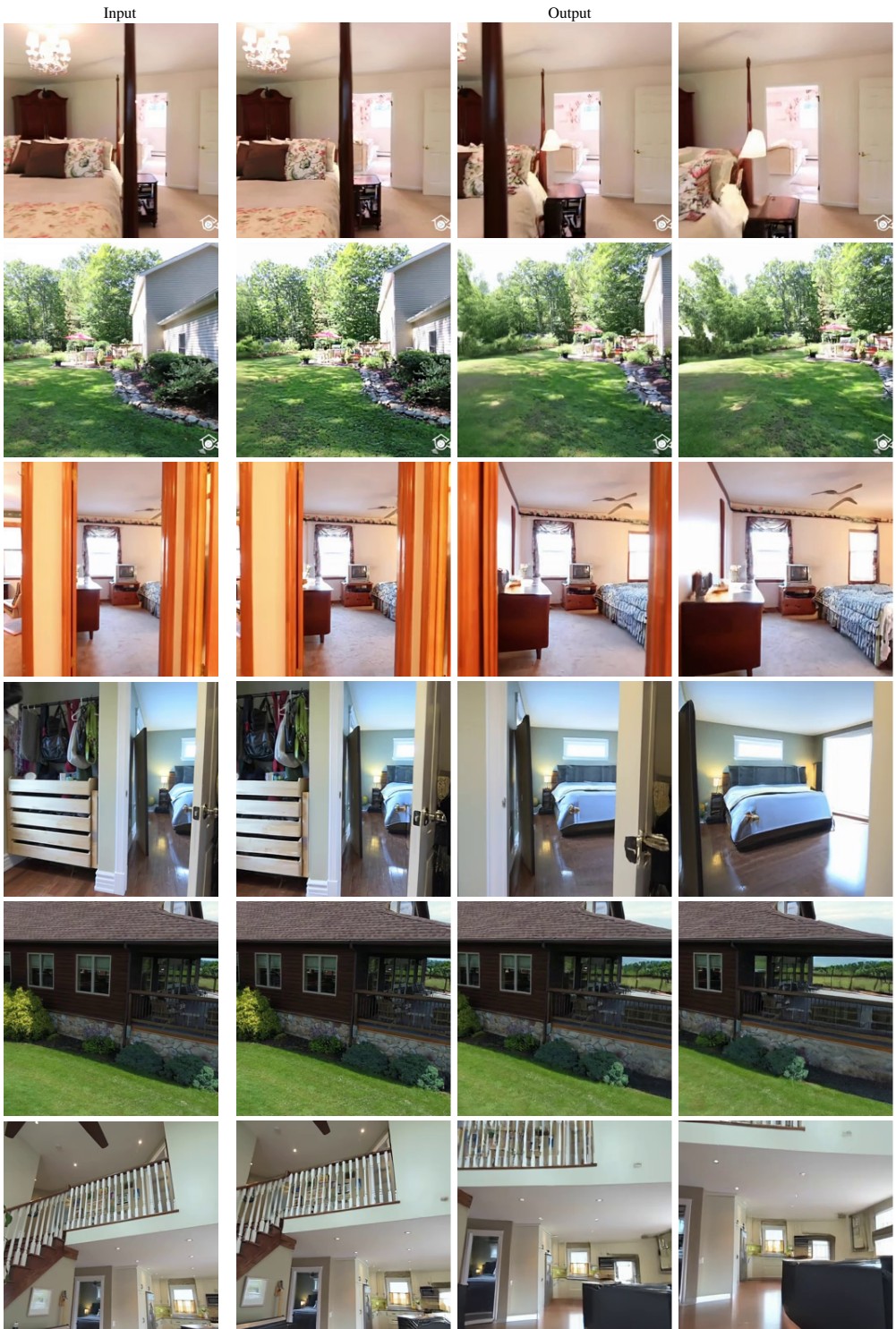

Figure 11: More 4DiM samples from the RealEstate10K dataset.

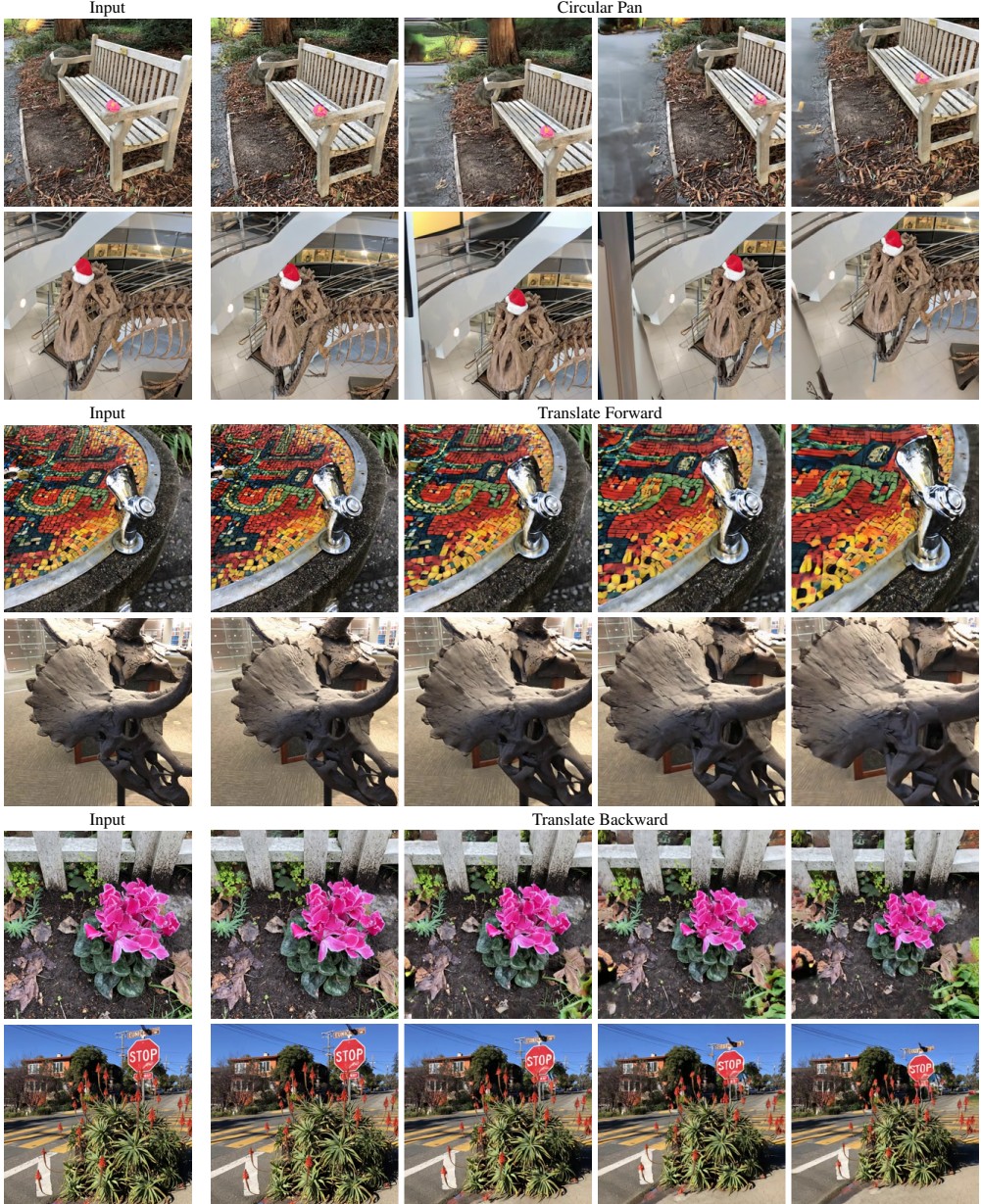

Figure 12: More 4DiM samples with custom trajectories from the LLFF dataset.

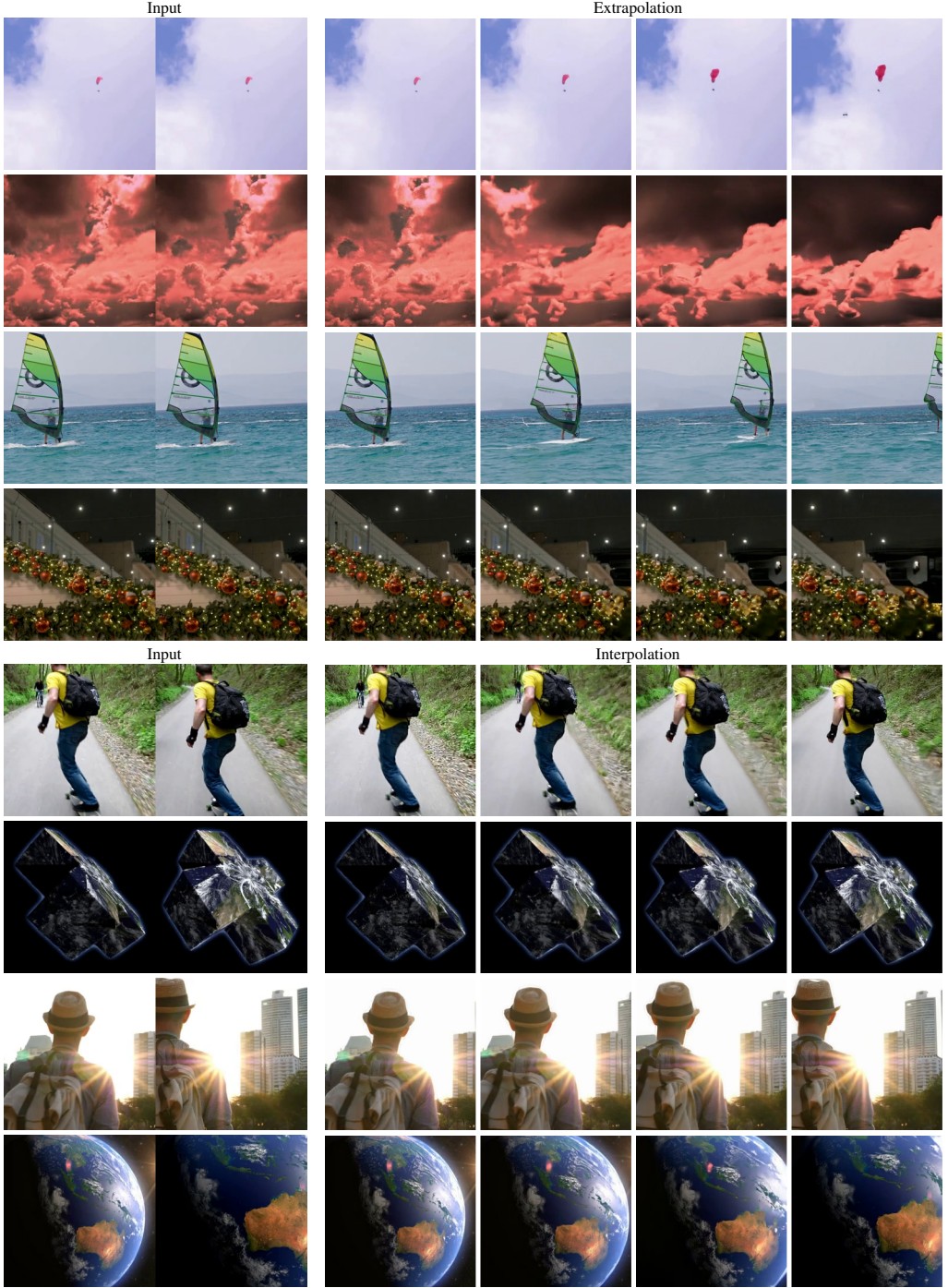

Figure 13: More 4DiM samples of videos generated from 2 input images.

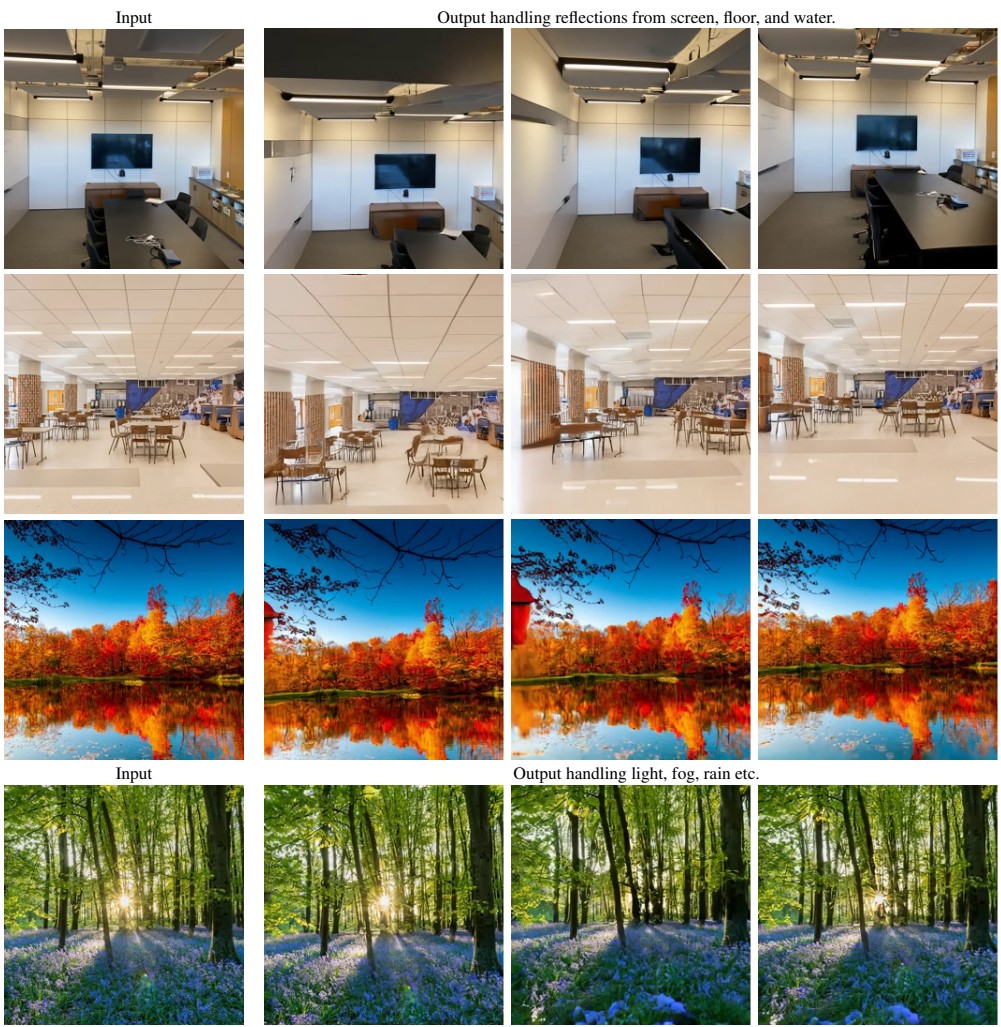

Figure 14: 4DiM can handle challenging scenarios such as reflection, light rays, fog, rain, etc.

