# OpenReview forum: "Controlling Space and Time with Diffusion Models"
_ICLR.cc/2025/Conference — ICLR 2025 Poster_

### Official Review · Reviewer_SN35 · 2024-10-27

**Soundness:** 2
**Presentation:** 2
**Contribution:** 3
**Rating:** 6
**Confidence:** 3

**Summary:**

The paper proposes a two-stage diffusion model that generates image sequences conditioned on camera pose and time. To address the lack of pose-aware image sequence datasets, the paper trains the model on a mixture of datasets (posed and unposed) with additional data preprocessing steps and a modified architecture. The qualitative results demonstrate that the model achieves better generalizability across both in-domain and out-of-domain scenes with better alignment with the input camera parameters.

**Strengths:**

1. The qualitative results look promising, showing generalizability to new scene types and arbitrary camera trajectories while outperforming the baselines.
2. The paper incorporates different sources of dataset to achieve generalizability of the model.
3. The paper proposes a model architecture, Masked FiLM, to incorporate CFG training with incomplete conditional inputs.
4. The evaluation metrics comprehensively assess various aspects of a pose-aware generative model.

**Weaknesses:**

1. In Tab.1, 4DiM shows discrepancy in reconstruction-based metrics (LPIPS, PSNR, SSIM) between in-domain (RE10K) and out-of-domain (LLFF) samples. While the authors mention that the reconstruction metrics favor the blurry images generated by the baseline method, this implies that the proposed method cannot accurately reconstruct images at specific camera poses. This can be seen as contradicting the claims in the main text, such as "precise camera control" and "accurate control over camera pose." Similar trend is observed in Tab.2. The reasoning and explanations for the results on out-of-domain data could be presented in more detail.

2. Training on a mixed dataset is not a novel approach and has been proposed in previous works on different domains: MiDaS for depth estimation and dust3r in 3D reconstruction. Highlighting the key differences could help the readers to better understand the novel contributions.

3. Comparisons to previous works on multi-frame conditioning could be strengthened. When more than a single input image is given, one can consider generating sequence of images using frame-to-video generative models (e.g., FILM) or autoregressive models (e.g., Show-O). Similarly, comparisons to previous works in image-to-panorama generation (e.g., MVDiffusion, PanoDiff) are not presented. The effectiveness of the proposed method would be more convincing with the addition of relevant baselines for each application.

4. The ablation study in Table 3 shows that the effect of co-training with video is not significant, and qualitative comparisons between the two cases are not presented ("see Supplementary Material H for more samples"). The authors could provide the more explanations for interpreting the ablation study.

Some missing relevant works: Multi-Guidance (e.g., Instruct-Pix2Pix, StyleCrafter), Training on a Mixture of Datasets (eg., MiDaS, dust3r).

**Questions:**

1. It appears that 4DiM is trained from scratch rather than fine-tuned on a pre-trained diffusion model. Is there a particular reason for this choice?

---

> ### Author Response · Authors · 2024-11-22
> **Reply to reviewer SN35**
>
> Thank you for your review. We address the reviewer’s concerns below:
>
> **Weakness 1 (metrics):** Reconstruction based metrics have several limitations with respect to evaluating the performance of generative models. This is particularly exacerbated for out-of-domain evaluation (e.g. on LLFF) where the generated samples are expected to diverge from the ground truth even more. Given the imperfections of various metrics, they must be analysed in aggregate to draw conclusions. In Tab 1, our method consistently outperforms baselines on sample fidelity (FID, FDD) and also in 3D consistency and pose alignment (TSED). A similar trend is observed in Tab 2 for our calibrated vs uncalibrated models. We are happy to discuss this further during the discussion.
>
>
> **Weakness 2 (mixing data):** Absolutely, we do not claim that mixing datasets is novel; on the dataset front, our main contribution is highlighting the value of training on *metric-scale* calibrated data and the demonstrated effectiveness of the specific mixture we use.
>
> **Weakness 3 (comparisons / 2-image-to-video):** Thank you for this valuable feedback. We have now included qualitative comparisons to FILM for the video interpolation task in our anonymized website. We additionally included other baselines, RIFE (non-generative), and diffusion baseline LDMVFI where the same samples were made available to us for comparison, as the reviewer also suggested a generative baseline) [1,2]. Our model is competitive with all of these baselines, and does a much better job when there is a lot of motion between the start and end frame, whereas non-generative baselines present severe artifacts in such cases. We include FID scores below:
>
> | Method | FID |
> |--|--|
> | RIFE | 57.68 |
> | FILM | 68.88 |
> | LDMVFI | 56.28 |
> | 4DiM | 52.89 |
>
> - [1] RIFE: Real-time intermediate flow estimation for video frame interpolation, Huang et al 2022
> - [2] LDMVFI: Video Frame Interpolation with Latent Diffusion Models, Danier et al 2023
>
> **Weakness 3 (comparisons/panorama):** As for comparisons with panorama generation methods, please note that our method is *not* text conditional, and the baselines the reviewer suggested are all text-conditional (in particular panodiff has a two-stage pipeline where the second stage is text-conditional). For a fair comparison with ours, we run MVDiffusion with an empty prompt (which was their unconditional setting during training for CFG on text). We find that text-unconditioned MVDiffusion performs poorly for panorama generation. We include a few qualitative comparisons in our anonymized website.
>
> **Weakness 4 (co-training video):** Thank you for this feedback. We have now included some side-by-side qualitative comparisons of our model trained with and without video data in our anonymized website to demonstrate that, without access to large-scale video data of the real world, these models do not achieve a good understanding of the geometry of objects and in-the-wild scene content.
>
> **Weakness 5 (related)** Thanks for pointing these out, we will include these in the final version, in particular that 2-guidance was originally proposed in InstructPix2Pix and that our contribution is the generalization to N variables and its effectiveness for 4D NVS. As for training on mixtures of datasets, please see our response to Weakness 2.
>
> As for the remaining question:
>
> **Question 1 (from-scratch vs. fine-tuning):** The rationale behind this is non-technical. Due to our institution's policies around potential safety risks with training data used in external generative models, we were unable leverage prevalent pre-trained models for our research.

---

> > ### Comment · Reviewer_SN35 · 2024-11-26
> >
> > The authors have addressed my concerns, and the work shows a promising direction in novel view synthesis.
> >
> > As mentioned in the response, one of the core contributions of this work is training on a multi-sourced dataset, and I believe the work will have greater impact to the community if the trained model and dataset are publicly released.
> >
> > I can further raise my score upon clarification regarding the reproducibility of the work.

---

> > > ### Author Response · Authors · 2024-12-03
> > > **Additional reply to reviewer SN35**
> > >
> > > Thank you for the response. We are committed to releasing as much detail as we can to support future research.
> > >
> > > On the data front, we are excited about sharing our calibrated RealEstate10K dataset and we are working on securing necessary permissions from our institution. The other 3D datasets of static scenes that we use (Matterport3D and ScanNet++) are already publicly available. For 4D, while we cannot re-distribute our exact StreetView split since it requires additional approvals, it is possible to collect such a dataset using the StreetView API directly. In the paper, we have included details of how we sample trajectories from StreetView for training.
> > >
> > > Although releasing the model may be hard due to our institution's policies, we have instead included details about our architecture and other training and inference hyperparameters in the main paper and in the supplementary.

---

### Official Review · Reviewer_ossm · 2024-10-27

**Soundness:** 2
**Presentation:** 3
**Contribution:** 3
**Rating:** 6
**Confidence:** 4

**Summary:**

The paper introduce a new video generative model that has both camera and time control. This is achieved by carefully assembling static/dynamic training data and a model design that help leverage missing condition signals (e.g., camera poses / time variation). The qualitative results are promising, showing good disentanglement between camera and time. The quantitative result is better than existing methods on RE10k/LLFF on novel view synthesis and camera controllability.

**Strengths:**

- They introduce masked FiLM layers, which avoids misusing 0 conditions (1) during conditioning signal dropout and (2) due to missing data.
- They introduce cRealEstate10K dataset to unify the scale of camera extrinsics, which reduces ambiguities during training and leads to more reliable camera control.
- The qualitative results are promising, showing good disentanglement between camera and time.
- The quantitative result is better than existing methods on RE10k/LLFF.

**Weaknesses:**

- It's unclear whether the problem should be called 4D NVS. One common interpretation of 4D is 3D+time. Under this interpretation, I don't think the proposed model can do 4D. With camera control, it can do 3D. With time control, it can do 2D+time. However, combining both is not equivalent to 3D+time. 3D+time would require generating multiple videos that are synchronized in time, or multiple "frozen" time videos with different t.
- Data: The model is trained on mostly static data. The only source with non-static content is 1000 Google street views. How diverse is the data, and how much dynamic contents are there?  Does it include human motion and scene motion (e.g., tree, cars)?
- Result/eval: The reported metrics are mostly for 3D NVS. The only metric related to dynamics is keypoint distance, which however, only assess how large the motions are, but not how good the motions are. Additionally, the evaluation protocol is not clear enough -- is camera parameters the same for all the frameds when generating the videos? Otherwise, the camera motion would produce large keypoint distance.
- Disentanglement. The ability of disentangling camera motion and scene dynamics is not fully tested. It would help to report metrics that quantitatively measures the disentanglement. For example, whether fixing camera and changing t would introduce background motion; whether fixing t and changing camera will introduce object motion (e.g., through epipolar geometry).
- Some claims can be supported by experiments.
  - The benefit of masked FiLM layers.
  - In the supplement, "Unlike Plucker coordinates, rays additionally preserve camera positions which is key to deal with occlusions in the underlying 3D scene."

**Questions:**

- The paper compared to MotionCtrl and found it has trouble following pose controls. Is there a hypothesis for this observation? Is it due to lack of metric scale training poses, or are there other underlying factors? What happens if MotionCtrl is trained on cRE10K that has metric scale poses?
- What is the difference between the proposed multi-guidance and InstructPix2Pix [B] Eq 3.
- Will the data/model be open-sourced?
- Related works. [A] can be discussed as the setup is close although they do not claim time control.

[A] VD3D: Taming Large Video Diffusion Transformers for 3D Camera Control
[B] InstructPix2Pix Learning to Follow Image Editing Instructions.

---

> ### Author Response · Authors · 2024-11-22
> **Reply to reviewer ossm**
>
> Thank you for your review. We address the reviewer’s concerns below:
>
>
> **Weakness 1 (terminology):** We agree with the 3D+time interpretation of 4D. We think that the video-to-video translation examples on our website do illustrate 4D capabilities since we are able to generate the given input scenes (with dynamics) along new camera trajectories. We agree that the fidelity of temporal dynamics can be further improved but that is to be expected given we are one of the first to target this very challenging problem and the limited scale of our models. We strongly believe that fidelity in dynamics can be improved with further advancements in training mixtures and scaling up the size of these models.
>
> **Weakness 2 (data):** Please note that the majority of our training data actually consists of unposed videos (30 million videos) which are rich in temporal dynamics and include motion of objects, people/animals, camera, etc., i.e. it is not true that the model is trained mostly on static data. Indeed, one of our main contributions is precisely how to achieve novel view synthesis combining camera and time control (which we have demonstrated e.g. in our video-to-video translation results originally included in the anonymized website) despite present limitations in the availability of 4D data.
>
> **Weakness 3 (keypoint distance):**
>
> While our proposed keypoint distance metric does not truly disambiguate camera and object motion, in practice, we find it useful for comparing the dynamics capabilities of models with good camera pose alignment (e.g. for tuning tgw in Fig 7 in supplementary) since it effectively serves as a measure of dynamics in that case.
> Please note that we also included FVD in the paper, which does correlate with the quality of temporal dynamics, i.e. keypoint distance was _not_ the only metric included to assess video quality.
>
> One question for the reviewer:
>
> > Additionally, the evaluation protocol is not clear enough -- is camera parameters the same for all the frames when generating the videos?
>
> We are not sure what is precisely being asked here; if the reviewer can clarify a bit more, we are happy to answer any remaining questions.
>
> **Weakness 4 (measuring disentanglement):**
>
> We agree that disentangled metrics for camera and object motion would be useful for evaluating 4D generative models and that our included metrics are limited in their ability to do this. However, to our knowledge, such metrics do not yet exist in literature, but we welcome any specific suggestions for metrics to try.
>
> Re `fixing camera and changing t`: If there existed a technique to reliably annotate static vs. dynamic masks for a generated video, one could then measure what the reviewer suggests. Unfortunately, research progress in motion segmentation has been limited to date, with results too inaccurate and noisy to attain a reliable metric. Qualitatively, we demonstrate in our video-to-video results that fixing the camera and changing t indeed results in rich motion.
>
> Re `fixing t and changing camera`: The epipolar metrics we report (TSED) should capture this. Did the reviewer have other metrics in mind?
>
>
> **Weakness 5 (FiLM / Plucker):** Thank you for the suggestion. We will attempt to include these ablations in the final version. These were too expensive to conduct in the rebuttal timeline against the models we originally trained for the paper submission.
>
>
>
>
> As for some of the remaining questions:
>
> **Question 1 (MotionCtrl):** The specific version of MotionCtrl that we use was trained on StableVideoDiffusion. We used this particular version because we needed a model that supports image conditioning (the prior version did not). However, since this was a follow-up of the original MotionCtrl, details on how this model was trained are not included in their paper. This makes it difficult to speculate on the potential causes for the reported performance. Unfortunately, finetuning this model on cRE10K was infeasible for us due to our institution's policy around potential safety risks in the training data used in the base model.
>
> **Question 2 (multi-guidance)**: Thank you for pointing this out, we will revise our paper to mention that 2-guidance had been previously proposed by InstructPix2Pix, which we here extend to N-variables and for a different task.
>
> **Question 3 (open sourcing):** We will release our novel calibrated RealEstate10K data and allow it for open research use, including the evaluation code. While we are additionally pushing to open-source the model, we cannot promise this due to institutional policies.
>
> **Question 4 (related work)**: Thank you for pointing out this work; we will be sure to update our literature review to additionally make note of VD3D and other recent/contemporaneous work.

---

> > ### Comment · Reviewer_ossm · 2024-11-26
> >
> > Thanks for the clarification. Many of my comments have been addressed.
> >
> > To follow up on my question of evaluation, how are the conditioning camera pose and time chosen under different evaluation protocols? I assume to measure 3D consistency, time conditioning is set to a constant value when camera pose changes; for dynamics, camera pose remains constant when time changes. It would be useful to elaborate and add those details to the paper.
> >
> >
> > > We think that the video-to-video translation examples on our website do illustrate 4D capabilities since we are able to generate the given input scenes (with dynamics) along new camera trajectories.
> >
> > The visual example v2v with camera control loos plausible but the geometry consistency **between videos** is not fully verified. One experiment would be running a multiview/stereo reconstruction method using view sets at every time instance. This would result in good quality dynamic 3D reconstruction if the generated videos are geometrically consistent with the input video.

---

> > > ### Author Response · Authors · 2024-12-03
> > > **Additional reply to reviewer ossm**
> > >
> > > Thank you for the additional response.
> > >
> > > > how are the conditioning camera pose and time chosen under different evaluation protocols
> > >
> > > For 3D consistency evals on RE10K, ScanNet++ and LLFF time is “masked” i.e. treated as unknown. Similarly for evaluating dynamics on videos, the camera pose is masked. We will clarify this in the paper.
> > >
> > > > running a multiview/stereo reconstruction method
> > >
> > > Thanks for this additional suggestion for further improving our evaluation setup, specifically regarding 3d consistency in the video-to-video case. Unfortunately we could not get this done in time for the rebuttal, but we will look into it for the final version of the paper.

---

### Official Review · Reviewer_m6rk · 2024-11-02

**Soundness:** 3
**Presentation:** 3
**Contribution:** 4
**Rating:** 8
**Confidence:** 5

**Summary:**

This paper proposes 4DiM, a cascaded diffusion model for 3D and 4D novel view synthesis. By a specially designed network architecture and sampling procedure, 4DiM enables joint training with posed and unposed video data, which results in better generalization and output fidelity. This also allows joint camera pose and time control in 4D novel view synthesis, making 4DiM a general model for various tasks like image-to-3D, two-image-to-video (interpolation and extrapolation), and pose-conditioned video-to-video translation. Qualitative and quantitative results show that the 4DiM performs favorably against priors art on static and dynamic scene generation.

**Strengths:**

S1: General model
4DiM enables joint training with posed and unposed video data, which not only improves the generalization and output fidelity, but also allows joint camera pose and time control in 4D novel view synthesis. It is shown to perform well on various tasks such as image-to-3D, two-image-to-video (interpolation and extrapolation), and pose-conditioned video-to-video translation.

S2: Better 3D consistency and pose alignment
Both qualitative and quantitative results show a consistent improvement over prior methods in terms of 3D consistency and camera pose alignment. Although it is a bit surprising that joint training with unposed video data leads to some worse metrics on in-domain data like cRE10k and ScanNet++ (Table 3).

S3: Good writing
The paper is well-written and easy to follow. The ablation studies also give good insights on where the performance gain comes from.

**Weaknesses:**

W1: Motion realism and temporal consistency
While the model shows great 3D consistency and pose alignment, the fidelity and temporal consistency of dynamic objects seems less impressive. In most dynamic scene results, the moving objects (cars, tires, animals, etc) either have unrealistic motion or temporal artifacts. This can probably be improved by including dynamic object data like Objaverse for training. It would also be good to show more results on single-image-to-4D.

W2: Cartoonish texture
Some results (especially in single-image-to-4D and long trajectory generation) have quite cartoonish texture.

**Questions:**

Q1: I’m wondering if the authors have any explanation for why adding unposed video data lead to worse performance on certain metrics (see S3).

Q2: Following W2, do the authors have any ideas on how to improve the texture fidelity in these cases?

---

> ### Author Response · Authors · 2024-11-22
> **Reply to reviewer m6rk**
>
> Thank you for your review. We address the reviewer’s concerns below:
>
> **Weakness 1:** Thank you for the interesting suggestion of using Objaverse / ObjaverseDy [1]. Using this dataset may improve results, but it’s uncertain whether the gains from training on an object-centric dataset will transfer to real-world scenes. An empirical study is definitely warranted. Other ideas like conditioning on the amount of motion, or motion filtering (c.f. Stable Video Diffusion), could also be used.
>
> -  [1] SV4D: Dynamic 3D Content Generation with Multi-Frame and Multi-View Consistency, Xie et al, 2024
>
> As for single-image-to-4D, it remains a challenging task as we note in the paper. Qualitative samples suffer from low fidelity in dynamics.
>
> **Weakness 2 + Question 2:** The loss of texture in long range generation is indeed quite surprising. One potential solution would be to leverage reward fine-tuning (e.g. [1]) to improve both image fidelity and detail. Supervised fine-tuning on a high-quality subset of our data (e.g. in [2]) may also be beneficial.
>
> -  [1] Directly Fine-Tuning Diffusion Models on Differentiable Rewards, Clark et al, 2024
> -  [2] Stable Video Diffusion: Scaling Latent Video Diffusion Models to Large Datasets, Blattmann et al, 2023
>
> **Question 1:** While fidelity (FID, FDD) improves across the board when adding video data, certain pose alignment and reconstruction metrics get worse. One possible explanation is that training on posed data alone should be expected to perform better on pose alignment than a diluted training mixture with both posed and unposed video. To help demonstrate the benefits of including large scale unposed video in our training mixture further, we provide qualitative samples of our models trained with and without video data in the rebuttal section of the website.

---

> > ### Comment · Reviewer_m6rk · 2024-11-26
> >
> > Thanks the authors for the rebuttal. Most of my comments and concerns are addressed.
> >
> > Although several limitations like unrealistic motion and cartoonish texture still exist in the synthesized videos, I believe that the proposed joint training scheme with video, 3D, and 4D data is a great first step towards generalizable 4D NVS, and that the community can benefit from this work upon release.
> >
> > As such, I keep my positive rating.

---

### Official Review · Reviewer_3Lp3 · 2024-11-03

**Soundness:** 3
**Presentation:** 3
**Contribution:** 2
**Rating:** 6
**Confidence:** 4

**Summary:**

The paper presents "4DiM", a novel cascaded diffusion model aimed at 4D novel view synthesis (NVS) and able to generate scenes that can be viewed from arbitrary camera trajectories and timestamps. The model envolves several techniques to achieve this goal, including a super resolution model, Masked FiLM layers and different CFG params for conditioning variables. The model is trained on a combination of posed and 3D, 4D and video data, including RealEstate10K calibrated with a monocular depth estimation model. The design of training dataset improve the fidelity of generation and generalization ability to unseen images and camera pose trajectories. The proposed model is evaluated on in-distribution and out-of-distribution datasets with multiple metrics, some of which are proposed by the authors.

**Strengths:**

The task of unposed 4D novel view synthesis is a challenging one, and the paper presents a novel approach to tackle it. Compared to previous works, the proposed model has two strengths:
- The sampling method is well designed. The multi-guidance sampling enables controls on images; camera poses, and timestamps.
- The dataset combination enhances the generalization ability of the model. The authors also give experimental results to explain the effectiveness of the dataset.
- The authors propose novel metrics such as keypoint distance to measure temporal dynamics and improved TSED for 3D consistency, which are well-suited to evaluate the complex requirements of 4D NVS.
Moreover, the ablation study is well-designed, and the results are well-explained. The paper is well-written and easy to follow.

**Weaknesses:**

Although the paper has several strengths, there are some weaknesses that need to be addressed:
- The sampling resolution is not high enough. Compared with SOTA methods, ViewCrafter generates videos with higher resolution (576x1024), while the proposed method only generates 256x256 resolution videos even with a super resolution model. Is it possible to generate higher resolution videos with shorter sequence length or smaller training batch size?
- The discussion of 360-degree videos is not enough. The model is not evaluated on 360-degree datasets, MipNeRF360 for example, which is a common task in NVS.

**Questions:**

There are some questions that need to be clarified:
- The sampling efficiency and cost of the proposed model is not clear. How long does it take to generate a video compared to other methods? How much GPU memory is needed?
- In appendix F, the authors mention the gray padding for MotionCtrl LLFF samples in Figure 3. I don't understand the purpose of "in order to preserve the aspect ratio of LLFF" since the sampling resolution is 567\*1008. Why is it necessary to add gray padding as the sampling resolution is already higher than 256\*256? Does the gray padding affect the metrics of the model?
- gw, tgw and pgw in appendix Figure 7 are not explained in the main text.

---

> ### Author Response · Authors · 2024-11-22
> **Reply to reviewer 3Lp3**
>
> Thank you for your review. We address the reviewer’s concerns below:
>
>
> **Weakness 1 (low res):** While higher resolutions can be achieved with pixel space diffusion models, they typically require deeper cascades [1, 2]. Latent space models provide a simpler and more compute-efficient alternative. Since the time of the original submission, we also trained a 512x512 latent-space 4DiM. We have now added some qualitative examples for 3D generation with this model in our anonymized website. As such, we emphasize that our findings in this submission are also applicable, “as is”, to higher resolution generation with latent diffusion models.
>
> - [1] Photorealistic Text-to-Image Diffusion Models with Deep Language Understanding, Saharia et al, Neurips 2022
> - [2] Imagen Video: High Definition Video Generation with Diffusion Models, Ho et al, 2022
>
> **Weakness 2 (MipNeRF360):** The MipNerf360 dataset is out-of-distribution for our training data; the object-centric scene content and trajectories (spins around objects) in MipNerf360 are very different from those in any of our training datasets. Hence an evaluation may not provide meaningful insights. With appropriate training data added to our mixture, we are confident we would obtain high quality results on object-centric benchmarks, however, we leave that to future work as our work focuses on scenes and demonstrating the effectiveness of training with metric scale-calibrated data.
>
> **Question 1 (efficiency):** Below we provide latency comparison of 4DiM and MotionCtrl.
> |              | 4DiM | MotionCtrl |
> |--------------|------|------------|
> | Walltime (s) | 125  | 142        |
>
> While we evaluated both methods with a batch size of 1, there are some differences worth noting. MotionCtrl (576x1024) has a larger spatial resolution than 4DiM (256x256). However, it only generates 16 frames whereas 4DiM generates 32 frames. Furthermore, 4DiM was evaluated on a TPU v3 with 16GB HBM whereas MotionCtrl on a NVIDIA A100.
>
> It is worth highlighting that with recent advances in efficient sampling of diffusion models, it is possible to run inference with much fewer denoising steps leading to much faster inference. However, we leave such optimizations to future work.
>
> **Question 2 (gray padding):** To enable fair comparisons across models and datasets with different aspect ratios (e.g. MotionCtrl uses 576x1024, 4DiM works on square images, while LLFF is at 756x1008) we chose to evaluate on center square crops. However, MotionCtrl, owing to a wider aspect ratio than LLFF, was evaluated on a vertically cropped LLFF (to match the desired aspect ratio). The gray padding illustrates the cropped out region and was not included when computing the metrics. Please let us know if anything remains unclear here.
>
> **Question 3 (notation):** Thanks for noticing this. We will amend the caption to make this more clear. More specifically, “gw” is standard “guidance weight”, while “tgw” stands for “timestamp”-gw, and “pgw” stands for “pose”-gw, where the latter refer to multi-guidance; varying the weight on these respective signals while keeping image guidance weight constant.

---

> > ### Comment · Reviewer_3Lp3 · 2024-11-26
> > **Thanks for the rebuttal.**
> >
> > Thank you for the detailed response. Most of my concerns have been addressed. This work explores video diffusion models' generative priors and achieves promising results. This approach may lead to a new era in scene generation. However, as noted by Reviewer SN35, the reproducibility of this work should be further clarified to support future research.

---

> > > ### Author Response · Authors · 2024-12-03
> > > **Additional reply to reviewer 3Lp3**
> > >
> > > Thank you for the additional response. Regarding reproducibility, please refer to our response to SN35.

---

### Author Response · Authors · 2024-11-22
**Reply to all reviewers (REQUESTED SAMPLES IN WEBSITE)**

We thank the reviewers for their feedback. We address their comments individually and provide additional qualitative results, requested by reviewers, in our anonymized website (with permission from AC): [https://anonymous-4d-diffusion.github.io/index.html#rebuttal](https://anonymous-4d-diffusion.github.io/index.html#rebuttal)

---

### Meta-Review · Area_Chair_monX · 2024-12-20

**Metareview:**

This paper presents a diffusion model that can generate novel views and/or generate motion. The key technical contribution is to unify these tasks in a common architecture using masked-Film layers that allow optional time and camera conditioning when synthesizing images, while also enabling training with unposed videos. The results demonstrate the ability of the model to synthesize novel views as well as perform video interpolation/extrapolation. The reviewers are generally positive about the contributions and results, and the AC agrees with this consensus. The main caveat is that the “4D” in the title/positioning is a bit misleading as the model can only generate a single space-time trajectory (as opposed to current ‘4D’ methods generating multi-view images across time), and the AC would request the authors to update the title with this consideration.

**Additional Comments On Reviewer Discussion:**

The initial ratings for this work were mixed/negative, but the author response assuaged the primary concerns, and the reviewers are finally all positive.

---

### Decision · Program_Chairs · 2025-01-22

Accept (Poster)